# Metabolic imaging across scales reveals distinct prostate cancer phenotypes

Nikita Sushentsev ⬤[1,13] ✉, Gregory Hamm[2,13], Lucy Flint[2], Daniel Birtles ⬤[2], Aleksandr Zakirov[3], Jack Richings[4], Stephanie Ling ⬤[2], Jennifer Y. Tan[4], Mary A. McLean ⬤[1,5], Vinay Ayyappan[1], Ines Horvat Menih[1], Cara Brodie[5], Jodi L. Miller ⬤[5], Ian G. Mills ⬤[6,7], Vincent J. Gnanapragasam[8,9,10], Anne Y. Warren ⬤[11], Simon T. Barry ⬤[12], Richard J. A. Goodwin ⬤[2,14], Tristan Barrett[1,14] & Ferdia A. Gallagher[1,14]

Hyperpolarised magnetic resonance imaging (HP-[13]C-MRI) has shown promise as a clinical tool for detecting and characterising prostate cancer. Here we use a range of spatially resolved histological techniques to identify the biological mechanisms underpinning differential [1-[13]C]lactate labelling between benign and malignant prostate, as well as in tumours containing cribriform and non-cribriform Gleason pattern 4 disease. Here we show that elevated hyperpolarised [1-[13]C]lactate signal in prostate cancer compared to the benign prostate is primarily driven by increased tumour epithelial cell density and vascularity, rather than differences in epithelial lactate concentration between tumour and normal. We also demonstrate that some tumours of the cribriform subtype may lack [1-[13]C]lactate labelling, which is explained by lower epithelial lactate dehydrogenase expression, higher mitochondrial pyruvate carrier density, and increased lipid abundance compared to lactate-rich non-cribriform lesions. These findings highlight the potential of combining spatial metabolic imaging tools across scales to identify clinically significant metabolic phenotypes in prostate cancer.

Hyperpolarised [1-[13]C]pyruvate MRI (HP-[13]C-MRI) is an emerging non-ionising metabolic imaging technique to probe tumour metabolism[1]. The method offers several advantages over routine [[18]F]2-fluoro-2-deoxy-D-glucose ([[18]F]FDG) positron emission tomography (PET), including discrimination of the injected substrate from its metabolic products to quantify cellular metabolism within specific tumour compartments such as the epithelium[2], as well as sensitivity to lesions with low glucose uptake including prostate cancer (PCa)[3]. Several clinical reports have shown the potential of HP-[13]C-MRI to detect PCa, assess disease aggressiveness, and detect tumour response to

[1]Department of Radiology, University of Cambridge and Cambridge University Hospitals NHS Foundation Trust, Cambridge, UK. [2]Integrated BioAnalysis, Clinical Pharmacology & Safety Sciences, R&D, AstraZeneca, Cambridge, UK. [3]Department of Clinical Neurosciences, University of Cambridge, Cambridge, UK. [4]Predictive AI & Data, Clinical Pharmacology & Safety Sciences, R&D, AstraZeneca, Cambridge, UK. [5]Cancer Research UK Cambridge Institute, University of Cambridge, Cambridge, UK. [6]Patrick G Johnston Centre for Cancer Research, Queen's University Belfast, Belfast, UK. [7]Nuffield Department of Surgical Sciences, University of Oxford, John Radcliffe Hospital, Oxford, UK. [8]Department of Urology, Cambridge University Hospitals NHS Foundation Trust, Cambridge, UK. [9]Division of Urology, Department of Surgery, University of Cambridge, Cambridge, UK. [10]Cambridge Urology Translational Research and Clinical Trials Office, Cambridge Biomedical Campus, Addenbrooke's Hospital, Cambridge, UK. [11]Department of Pathology, Cambridge University Hospitals NHS Foundation Trust, Cambridge, UK. [12]Bioscience, Early Oncology, AstraZeneca, Cambridge, UK. [13]These authors contributed equally: Nikita Sushentsev, Gregory Hamm. [14]These authors jointly supervised this work: Richard J.A. Goodwin, Tristan Barrett, Ferdia A. Gallagher. ✉ e-mail: ns784@medschl.cam.ac.uk

therapy[2–7]. Importantly, some of these studies have attempted to explain the mechanism of the observed tumour [1-$^{13}$C]lactate labelling patterns by correlating imaging data with tissue-based metabolic biomarkers[2,8–10]. However, a more comprehensive biological validation of HP-$^{13}$C-MRI is required to assess its translational potential and navigate its use in specific clinical scenarios[11].

In addition to detection of PCa by differentiating it from the healthy prostate, an area of particular unmet need is the non-invasive phenotyping of intermediate-risk disease[12], which includes tumours comprised of Gleason pattern 4 (GP4) glands of cribriform and non-cribriform morphology. In 2019, both the International Society of Urological Pathology (ISUP) and Genitourinary Pathology Society (GUPS) recommended the reporting of invasive cribriform carcinoma (ICC) in biopsy specimens due to its higher recurrence rate and increased PCa-specific mortality compared to non-cribriform lesions[13,14]. While recent studies in this area have mostly focused on transcriptomic features of ICC[15–17], little is known about its metabolic properties in contrast to other PCa phenotypes. In addition, the appearances of this tumour variant on HP-$^{13}$C-MRI are unknown, which is of high translational relevance since up to 80% of pure cribriform tumours can be missed on conventional MRI[18].

The current understanding of the biological determinants of [1-$^{13}$C]lactate labelling detected with HP-$^{13}$C-MRI is that it reflects a complex interplay between biological factors which are summarised in Fig. 1. The primary determinant is hyperpolarised [1-$^{13}$C]pyruvate delivery to the tissue of interest, which is a function of perfusion, microvascular density, and endothelial permeability[19–21]. The second major factor is the cellular capacity for [1-$^{13}$C]pyruvate uptake, which is primarily mediated by the monocarboxylate transporter 1 (MCT1)[4,22–24]. The third factor relates to the intracellular metabolic fate of [1-$^{13}$C] pyruvate, which is a function of its enzymatic exchange into [1-$^{13}$C] lactate, and mitochondrial flux as measured through the formation of $^{13}$C-bicarbonate[25]. The fourth factor represents the natural abundance of the endogenous intracellular lactate pool which accepts the $^{13}$C label from the imported [1-$^{13}$C]pyruvate[26]: previous clinical studies have inferred this indirectly from lactate dehydrogenase (LDH) expression[2,5] but it can also be measured directly using tissue-based spatial metabolomics within specific cellular regions of the tumour[27]. Finally, the fifth factor is the capacity of the tissue-of-interest to generate a sufficient [1-$^{13}$C]lactate signal-to-noise ratio (SNR) for detection with HP-$^{13}$C-MRI, which in part is influenced by the density of MCT1-expressing and lactate-abundant cells, as well as the extracellular lactate accumulation in the voxel of interest.

Here, we applied this five-factor biological framework to prospectively collect imaging and spatially resolved multi-modal data from two matched intermediate-risk PCa patient cohorts. We show that tumour [1-$^{13}$C]lactate labelling is primarily driven by the increased epithelial cell density and vascularity of PCa areas compared to the benign prostate, while the endogenous epithelial lactate concentration is similar between the two tissue types. However, we also demonstrate that some cribriform tumours lack [1-$^{13}$C]lactate signal on HP-$^{13}$C-MRI despite sufficient cellularity and vascularity, a finding potentially explained by their lower LDH expression and higher mitochondrial pyruvate flux compared to equally cellular and well-perfused non-cribriform lesions. Using spatial metabolomics acquired from mass spectrometry imaging (MSI), we then show that the epithelium in ICC has a low endogenous lactate pool in the presence of abundant unsaturated fatty acids, presenting a distinct metabolic phenotype compared to lactate-rich non-cribriform GP4 glands. These findings indicate that although [1-$^{13}$C]lactate labelling may be a powerful tool for non-invasively imaging tumour metabolism, it may provide negative contrast for ICC detection, representing a limitation for the characterisation of intermediate-risk PCa. This study also highlights the potential of tissue-based spatial metabolomics to guide future developments in clinical metabolic imaging

techniques that are specific to the metabolic features of selected tumours.

## Results

As described in Fig. 1, this study included two prospective cohorts of PCa patients who underwent robot-assisted radical prostatectomy (RARP) in our centre. The HP-$^{13}$C-MRI cohort included 8 patients (median age, 65 years) who underwent successful HP-$^{13}$C-MRI prior to RARP and represented a subset of a previously described patient population[2]. Histopathological examination of formalin-fixed paraffin-embedded (FFPE) whole-mount slides revealed the presence of 15 lesions, and the detailed morphological characteristics of these are presented in Table 1.

To complement the use of tissue-based biomarkers for inferring cellular metabolic phenotype, we sought to directly assess endogenous epithelial metabolite abundance by means of spatially resolved desorption electrospray ionisation mass spectrometry imaging (DESI-MSI). To avoid the potentially detrimental effect of using FFPE with DESI-MSI[28,29], fresh-frozen RARP samples from a cohort of 13 patients (median age, 64 years) were used for this analysis (spatial metabolomics cohort in Fig. 1). A total of 117 tissue cores were obtained, of which 61 and 56 were derived from 15 tumours and 15 benign tissue areas, respectively. For comparability, tumours from the two cohorts were matched for the key histopathological parameters presented in Table 1.

### Increased tumour epithelial cell density, not lactate abundance, explains the ability of HP-$^{13}$C-MRI to distinguish PCa from the benign prostate

Our first step in the biological validation of HP-$^{13}$C-MRI was aimed at dissecting the specific mechanisms behind its ability to detect PCa by distinguishing it from the healthy prostate. In the HP-$^{13}$C-MRI cohort, both hyperpolarised [1-$^{13}$C]pyruvate and [1-$^{13}$C]lactate were measured in histopathologically-proven tumour areas ($n = 13$) ($P < 0.001$ for both; Fig. 2b) which corresponded to local metabolic hotspots on $k_{PL}$ maps ($P < 0.001$; Supplementary Fig. 1a) and were in line with similar findings described in prior clinical reports[3,4,30]. A similar trend was noted in the DESI-MSI cohort, where the whole-core measurements of absolute lactate abundance were significantly higher in tumour samples compared to benign specimens ($P < 0.0001$; Fig. 2c), in keeping with prior studies using bulk MSI[31,32]. However, these results can be partially explained by tissue density, which is known to increase in PCa compared to the benign tissue due to the obstruction of empty glandular spaces by neoplastic epithelium[33]. Hence, when the whole-core lactate abundance was corrected for tissue density, which was significantly higher in tumour samples ($P = 0.007$; Fig. 2c), the difference in endogenous lactate between the two tissue types reduced but remained significant ($P = 0.043$; Fig. 2c).

Given the established metabolic compartmentalisation within the human prostate[2,34–36], it was then important to identify the cellular source for the increased lactate abundance in PCa samples. We first trained a DenseNet model to segment each tissue section into stromal and epithelial compartments, allowing their proportions to be quantified in our specimens. As expected in this age group[37], benign cores harboured a significantly higher proportion of stromal tissue, whereas tumour samples were predominantly epithelial ($P < 0.01$ for both; Fig. 2d). While the epithelial lactate abundance was similar between benign and tumour cores ($P = 0.54$; Fig. 2e and Supplementary Fig. 2a for patient-to-patient variation), it was the significant increase in the stromal lactate ($P < 0.0001$; Fig. 2e) that explained the difference in whole-core lactate measurements observed in Fig. 2c. However, despite the high concentration of lactate in the stroma, this is unlikely to make a significant contribution to the imaged HP-$^{13}$C-MRI signal due to negligible MCT1 expression[2] and therefore slow pyruvate uptake. Consequently, the imaged tumour [1-$^{13}$C]lactate labelling noted in the

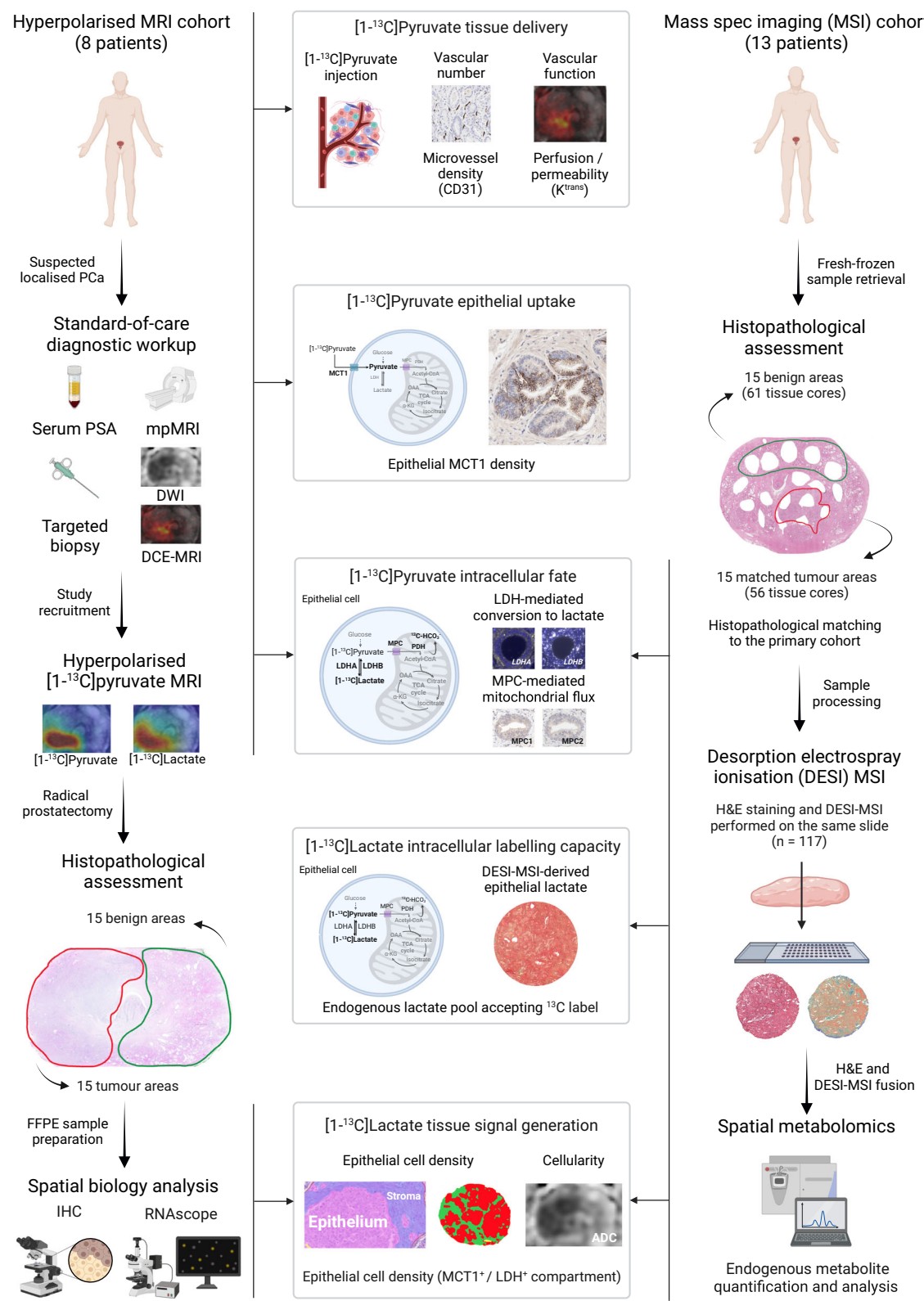

**Fig. 1 | Study design.** This clinical study included two prospective surgical cohorts of PCa patients whose imaging data and surgical specimens were analysed to measure five biological factors that can influence clinical [1-¹³C]lactate labelling. In the Hyperpolarised MRI cohort (n = 8 patients), quantitative ¹H-MRI, immunohistochemistry (IHC), and RNAscope data were used to infer tissue delivery, epithelial uptake, and the intracellular metabolic fate of [1-¹³C]pyruvate in 15 tumour and 15 benign areas. This was complemented by spatial metabolomic analysis of a set of histologically matched fresh-frozen benign (n = 61) and tumour (n = 56) samples from a DESI-MSI cohort (n = 13 patients), which enabled us to assess the endogenous lactate pool as a measure of the cellular capacity for [1-¹³C]labelling. Digital pathology data from both cohorts were used to quantify the density of epithelial cells as a measure of tissue capacity for generating detectable [1-¹³C]lactate signal. Tumour characteristics and methods are detailed in Table 1 and Methods. This figure was created with BioRender.com and released under a Creative Commons Attribution-NonCommercial-NoDerivs 4.0 International license.

**Table 1 | Key histopathological characteristics of tumours included in this study**

| Hyperpolarised MRI cohort | | | | | | |
|---|---|---|---|---|---|---|
| Patient | PSA, ng/ml | Tumour | Final ISUP grade group | Final %GP4 | Dominant GP4 subtype | Final pT stage |
| 1 | 8.5 | 1 | 2 | <5 | Non-cribriform | pT3a |
| | | 2 | 2 | <5 | Non-cribriform | |
| 2 | 19.1 | 3 | 2 | 30 | Non-cribriform | pT2 |
| | | 4 | 2 | 20 | Non-cribriform | |
| 3 | 4.7 | 5 | 2 | 5 | Non-cribriform | pT2 |
| | | 6 | 1 | 0 | - | |
| 4 | 13.9 | 7 | 2 | 15 | Non-cribriform | pT2 |
| 5 | 3.1 | 8[a] | 3 | 50 | Non-cribriform | pT3a |
| | | 9[a] | 1 | 0 | - | |
| 6 | 12.5 | 10 | 2 | <5 | Non-cribriform | pT3b |
| | | 11[b] | 3 | 60 | ICC | |
| 7 | 7.6 | 12 | 2 | <5 | Non-cribriform | pT3a |
| | | 13[b] | 3 | 60 | ICC | |
| 8 | 6.6 | 14 | 2 | 10 | Non-cribriform | pT3a |
| | | 15 | 3 | 60 | ICC | |
| Spatial metabolomics cohort | | | | | | |
| 1 | 9 | 1 | 2 | <5 | Non-cribriform | pT2 |
| 2 | 4.5 | 2 | 2 | <5 | Non-cribriform | pT3a |
| 3 | 2.5 | 3 | 2 | 30 | Non-cribriform | pT2 |
| 4 | 17.2 | 4 | 2 | 20 | Non-cribriform | pT3a |
| 5 | 14 | 5 | 2 | 5 | Non-cribriform | pT2 |
| 6 | 3.4 | 6 | 1 | 0 | - | pT2 |
| 7 | 16.6 | 7 | 2 | 15 | Non-cribriform | pT2 |
| 8 | 7.2 | 8 | 3 | 50 | Non-cribriform | pT3a |
| 9 | 9.9 | 9 | 2 | <5 | Non-cribriform | pT2 |
| 10 | 6.6 | 10 | 2 | <5 | Non-cribriform | pT2 |
| 11 | 16.5 | 11 | 3 | 50 | Non-cribriform | pT3a |
| | | 12 | 3 | 60 | ICC | pT3b |
| 12 | 6.2 | 13 | 1 | 0 | - | pT3a |
| | | 14 | 3 | 60 | ICC | |
| 13 | 9 | 15 | 3 | 60 | ICC | pT3a |

[a]These lesions were excluded from the tissue-based analysis due to interval ADT prior to RARP[2].
[b]These lesions were HP-[13]C-MRI occult, with [1-[13]C]lactate SNR < 5.0 in both cases.

HP-[13]C-MRI cohort is likely to be heavily determined by the increased epithelial cell density within the malignant regions. In the HP-[13]C-MRI cohort, tumour epithelial cell density was significantly increased compared to the benign areas ($P = 0.1$; Fig. 2f) and was positively correlated with [1-[13]C]lactate SNR ($\rho_s = 0.76$; $P = 0.1$; Supplementary Fig. 3). On standard-of-care [1]H-MRI, this high tumour epithelial cell density was reflected by a significantly decreased tumour apparent diffusion coefficient (ADC) on diffusion-weighted imaging ($P < 0.0001$; Fig. 2g), with a strong negative correlation observed between the two parameters ($\rho_s = -0.67$; $P = 0.0006$; Fig. 2h). Given the previously reported negative correlation between tumour ADC and [1-[13]C]lactate SNR[2], these data further support the hypothesis that increased epithelial cell density is an important biological driver of increased [1-[13]C]lactate labelling in PCa compared to the benign prostate.

Another potential driver of tissue [1-[13]C]lactate labelling is the delivery of hyperpolarised [1-[13]C]pyruvate to the organ of interest. Tumours in the HP-[13]C-MRI cohort had a significantly higher CD31-derived microvessel density (MVD) on immunohistochemistry (IHC; $P = 0.034$; Fig. 2i), which was associated with a significant increase in tumour-derived $K^{trans}$ ($P = 0.003$; Fig. 2j), a [1]H-MRI-derived measure of vascular perfusion and permeability acquired after the intravenous administration of contrast media[38]. $K^{trans}$ and CD31 showed a moderate positive correlation between each other ($\rho_s = 0.55$, $P = 0.02$; Fig. 2k), which is in agreement with the established link between vascularity and contrast agent delivery to the tissue-of-interest[39]. Neither of these markers showed a significant relationship with tumour [1-[13]C]pyruvate or [1-[13]C]lactate SNR[2] (Supplementary Fig. 4), possibly due to the relatively homogeneous vascularity of low- and intermediate-risk tumours in this cohort[2]. However, increased tumour vascularity may further contribute to the [1-[13]C]lactate labelling identified in the malignant compared to the benign prostate.

Finally, given the importance of MCT1 for cellular [1-[13]C]pyruvate uptake and subsequent [1-[13]C]lactate labelling[4,22–24], we compared its expression between the benign and tumour samples in the HP-[13]C-MRI cohort. While MCT1 staining was predominantly epithelial[2], it was also heterogeneously distributed, with the resulting epithelial MCT1 density being similar between the benign and tumour areas ($P = 0.3$; Fig. 2l). While this agrees with some prior studies investigating prostate MCT1 expression using IHC[40,41], the TCGA-PRAD derived mRNA data[42] showed a significant overexpression of *SLC16A1* (MCT1) in tumour samples compared to benign surgical specimens ($P = 0.009$; Fig. 2m). The differences between the IHC and TCGA-PRAD results could be explained by limitations in the sample size of the former, and discrepancies between mRNA and protein expression which are well

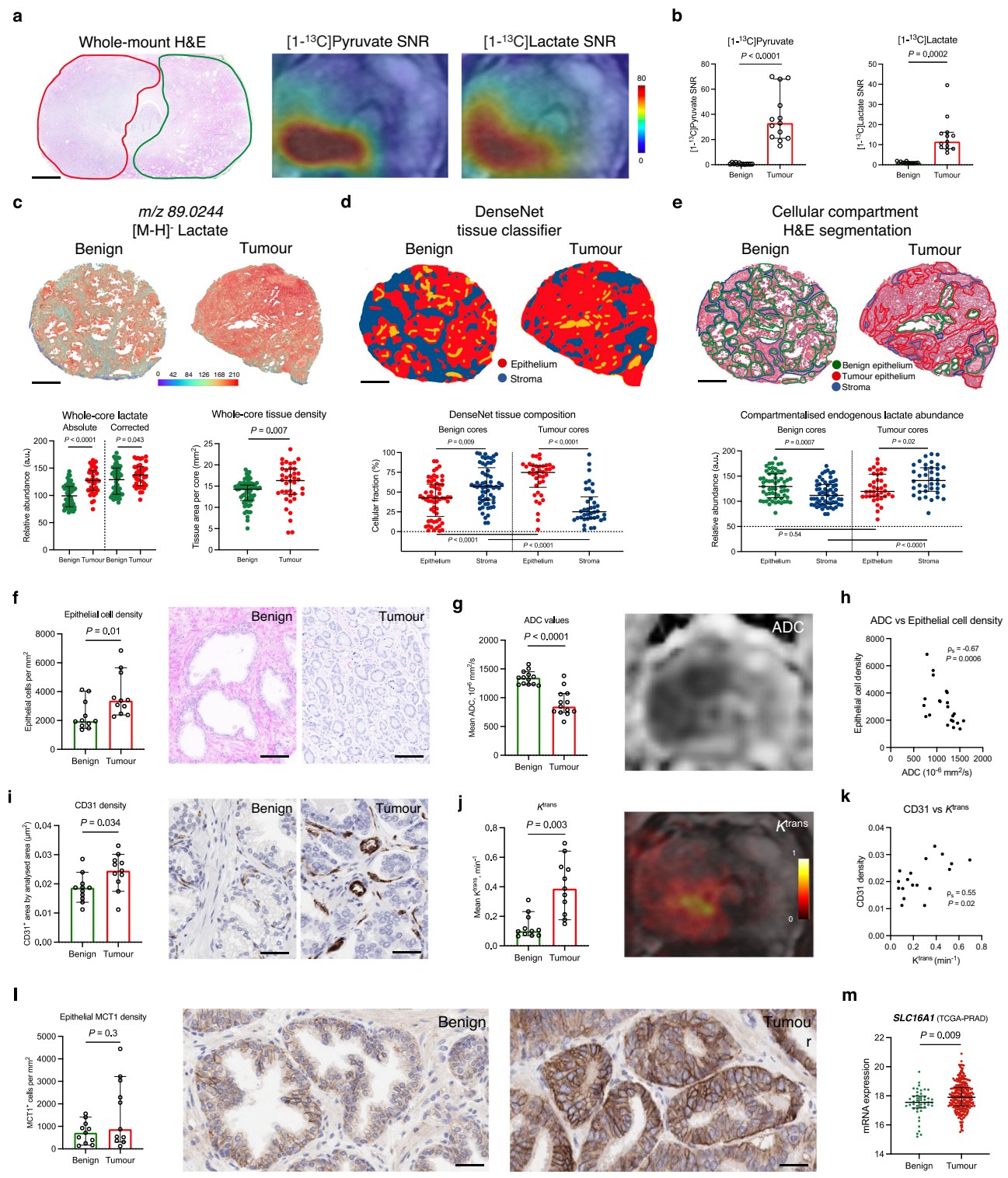

established[43]. Nonetheless, as no correlation between tumour MCT1 expression and [1-$^{13}$C]lactate SNR was noted in the HP-$^{13}$C-MRI cohort[2], epithelial MCT1 expression is unlikely to explain the increased tumour [1-$^{13}$C]lactate labelling observed in this study.

### Increased pyruvate mitochondrial import supports PCa metabolic reprogramming and correlates negatively with clinical [1-$^{13}$C]lactate labelling

Having dissected the potential influence of some of the aforementioned biological factors on differential [1-$^{13}$C]lactate labelling between the benign and malignant prostate, we then explored the dominant intracellular metabolic fate of [1-$^{13}$C]pyruvate in the two tissue types. As shown in Fig. 3a, glycolytic pyruvate formation is followed by either its enzymatic conversion to lactate or mitochondrial flux resulting in the synthesis of acetyl-CoA, which undergoes a condensation reaction with oxaloacetate to form mitochondrial citrate. In the benign prostatic epithelium, further citrate oxidation within the tricarboxylic acid (TCA) cycle is truncated, with the accumulating citrate transported outside the cell through its mitochondrial and plasma membrane transporters[44,45]. Conversely, early-stage PCa shows fully restored TCA cycle, which has the capacity to generate sufficient NADH for mitochondrial ATP production[44,46]. Instead of being exported outside the

**Fig. 2 | Biological validation of the ability of HP-¹³C-MRI to detect PCa by differentiating it from the healthy prostate. a** Representative whole-mount H&E section of a surgical FFPE sample obtained from a patient who harboured a single focus of ISUP GG2 PCa (red outline) that showed increased [1-¹³C]pyruvate and [1-¹³C]lactate signal on corresponding HP-¹³C-MRI maps compared to contralateral benign prostate (green outline). **b** Plots comparing [1-¹³C]pyruvate and [1-¹³C]lactate SNR derived from the ROIs encompassing areas of HP-¹³C-MRI-visible PCa (n = 13 samples from n = 8 patients) and contralateral benign prostate (n = 13 samples from n = 8 patients) from the hyperpolarised MRI cohort. **c** Representative fused H&E and DESI-MSI lactate maps of benign and tumour FF cores from the spatial metabolomics cohort. Plots on the left compare the absolute and tissue-density-corrected whole-core DESI-MSI derived lactate abundance between the benign (n = 61) and tumour (n = 38) cores; plots on the right compare the whole-core tissue density between the two specimen types. **d** DenseNet tissue classifier outputs overlaid on the H&E images of representative benign (n = 61) and tumour (n = 38) cores, with the below plots comparing epithelial and stromal cell fractions both within and across the benign and tumour cores. **e** H&E maps of representative benign (n = 61 samples from n = 13 patients) and tumour (n = 38 samples from n = 13 patients) cores with manually segmented areas of benign epithelium (green), tumour epithelium (red), and stroma (blue) used to derive DESI-MSI measured of endogenous lactate abundance compared in the below plots between and across benign and tumour cores. Plots and representative images comparing epithelial cell density (**f**), ¹H-MRI-derived ADC (**g**), CD31 density (**i**), ¹H-MRI-derived $K^{trans}$ (**j**), and epithelial MCT1 density (**l**) between the benign (n = 11) and tumour (n = 11) areas from the hyperpolarised MRI cohorts. Present Spearman's correlation plots comparing ADC values with epithelial cell density (**h**), as well as CD31 with $K^{trans}$ (**k**) derived from both benign and tumour areas. **m** Plot comparing TCGA-PRAD derived *SLC16A1* mRNA expression between the benign (n = 45) and tumour (n = 293) prostatectomy samples from n = 293 patients. All plots are scatterplots with bars (lines are median values, bars are interquartile ranges) with *P* derived using the two-sided Mann–Whitney *U* test or Wilcoxon signed-rank test, as appropriate. Scale bars in **a**, **c–e**, and **f–l** denote 5 mm, 1 mm, and 50 μm, respectively. Source data are provided as a Source Data file.

cell, cytosolic tumour citrate is cleaved by ATP citrate lyase to form acetyl-CoA and fuel lipogenesis, to sustain membrane biosynthesis in the proliferating cells[47–50]. The molecules involved in the rate-limiting reactions of this metabolic reprogramming include the mitochondrial pyruvate carriers (MPCs), mitochondrial pyruvate dehydrogenase (PDH), and cytosolic fatty acid synthase (FASN)[47–50], and their expression is directly induced by the androgen receptor (AR). Given the similar lactate concentrations in benign and malignant prostate tissue, we hypothesised that the key change in the tumour intracellular pyruvate fate is its increased mitochondrial import, rather than cytosolic reduction to lactate.

To test this hypothesis, a random forest tissue classifier[2,51] was applied to the same segmented benign and tumour ROIs on histopathology used for HP-¹³C-MRI data extraction (Fig. 2a): total epithelial LDH mRNA density was derived, which we have previously shown to correlate strongly with tumour [1-¹³C]lactate labelling in the primary cohort[2]. We found that this total epithelial LDH density was significantly reduced in tumour ROIs compared to the benign areas (*P* = 0.024; Fig. 3b, c), which is in keeping with findings derived from open-source TCGA-PRAD bulk mRNA sequencing data[42] (Fig. 3e), the epithelial single-cell mRNA sequencing data obtained from two recent publications by Tuong et al.[52] and Song et al.[53] (Fig. 3f), and spatial transcriptomics data from a study by Erickson et al.[54] (Fig. 3g). While this did not have an apparent effect on the intracellular lactate pool, a reduction in the total LDH expression in tumour epithelium may be functionally consistent with its increased capacity for mitochondrial [1-¹³C]pyruvate import.

To evaluate the role of mitochondrial import, we used IHC to measure the expression of MPC1 and MPC2, with the latter previously shown to be significantly overexpressed in the human prostate under the direct transcriptional control of nuclear AR[49]. Here, we show almost exclusive epithelial MPC1 and MPC2 localisation in both benign and malignant prostate (*P* < 0.05 for both; Fig. 3d), with tumour epithelial MPC2 density being significantly higher compared to the healthy tissue (*P* = 0.005; Fig. 3b, c), an observation also supported by bulk[42], single-cell[52,53], and spatial[54] mRNA sequencing data presented in Fig. 3e–g. Importantly, tumour epithelial MPC2 density showed a strong negative correlation with clinical [1-¹³C]lactate labelling ($\rho_s$ = −0.80, *P* = 0.005; Fig. 3d), which aligns with a prior preclinical report showing an increase in glycolytic intermediates following MPC inhibition[55]. Taken together with our previous work which showed strong positive associations between tumour [1-¹³C] lactate SNR, total epithelial LDH density, and epithelial *LDHA/PDHA1* ratio[2], these results indicate that tumour [1-¹³C]lactate labelling is likely to be a function of both the epithelial capacity for cytosolic LDH-catalysed pyruvate reduction, and MPC-driven mitochondrial pyruvate import.

We then explored the potential role of MPC2 in supporting AR-driven tumour reprogramming towards a lipogenic phenotype, by measuring the expression of both AR and FASN in our samples. As expected, tumour areas exhibited a significant increase in epithelial FASN density (*P* = 0.003; Fig. 3b, c; supported by Fig. 3e–g), which correlated strongly with the epithelial MPC2 density ($\rho_s$ = 0.93, *P* = 0.001; Fig. 3d). Both MPC2 and FASN also showed strong positive correlations with epithelial nuclear AR density ($\rho_s$ = 0.89 and 0.68, respectively; *P* < 0.05 for both; Fig. 3d), corroborating prior preclinical reports[49,50] identifying MPC2 and FASN as AR transcriptional targets. Overall, these findings suggest the potential role of mitochondrial pyruvate flux for enabling PCa metabolic rewiring and highlight its importance for clinical HP-¹³C-MRI interpretation.

## Spatial metabolomics combined with a deep learning metabolic classifier suggest differences in pyruvate metabolism between glycolytic benign prostate and lipogenic cancer

To further assess the role of mitochondrial pyruvate import in supporting lipogenic PCa metabolic reprogramming, we interrogated the DESI-MSI data derived from benign (n = 695) and tumour (n = 468) epithelial ROIs drawn manually on haematoxylin-and-eosin (H&E) slides obtained from the DESI-MSI cohort (Figs. 2e and 4e). We used the ROIs to extract DESI-MSI-derived metabolites from three relevant pathways within the KEGG database: glycolysis/gluconeogenesis, TCA cycle, and fatty acid biosynthesis pathways (Supplementary Table 1). Subsequent metabolic pathway enrichment analysis (MPEA) showed significant enrichment of the TCA cycle pathway in the benign ROIs (*P* = 0.01; Fig. 4a), which at the level of individual metabolites was primarily driven by citrate and phosphoenolpyruvate (PEP; *P* < 0.001 for both; Fig. 4b, c). Significant citrate enrichment in the benign prostate is well established[44,46] and therefore serves as a validation of the DESI-MSI approach. In turn, PEP enrichment in the benign tissue may be explained by both its glycolytic production and anaplerotic synthesis from cytosolic oxaloacetate, a process aimed at regenerating the TCA cycle carbons[56]. In addition, benign prostatic epithelium showed a non-significant enrichment of the glycolysis/gluconeogenesis pathway relative to PCa (*P* = 0.27; Fig. 4a), with PEP and glucose being significantly different (*P* < 0.0001; Fig. 4b, c). As in the case of citrate, the observed significant enrichment of glucose in the benign epithelium compared to intermediate-risk organ-confined PCa aligns with the published literature and explains the well-known limited diagnostic utility of [¹⁸F]FDG-PET in PCa[57]. Importantly, of the three pathways analysed, only metabolites involved in fatty acid biosynthesis were significantly enriched in tumour epithelial ROIs (*P* < 0.0001; Fig. 4a), including tetradecanoic, oleic, and palmitoleic acids being the three metabolite hits (*P* < 0.0001 for all; Fig. 4b, c). Notably, another significantly enriched tumour epithelial metabolite was S-

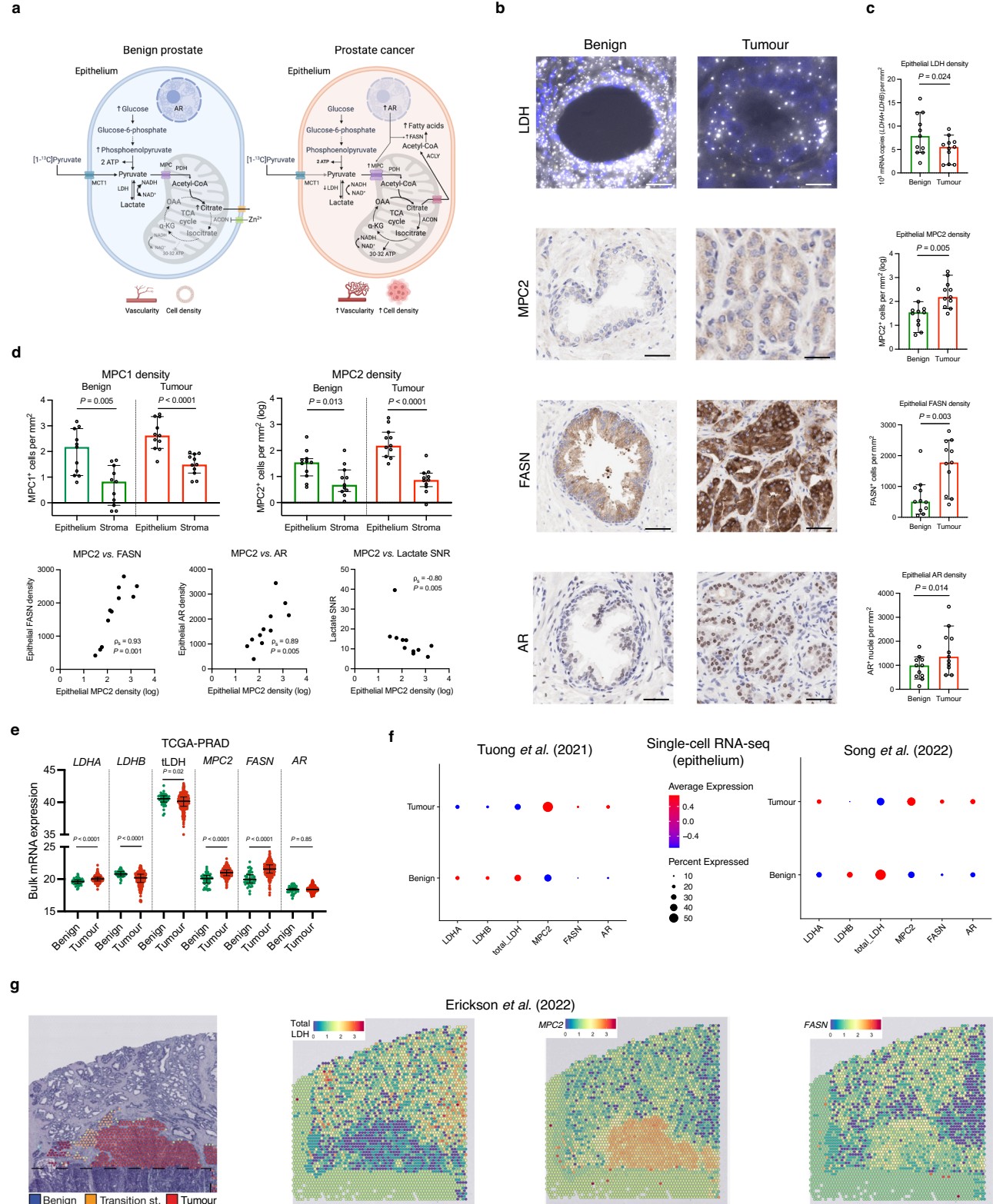

Acetyldihydrolipoamide-E, which is a pyruvate-derived precursor of mitochondrial acetyl-CoA formed as part of the PDH-catalysed reaction[58]. Increased tumour epithelial S-Acetyldihydrolipoamide-E enrichment supports the notion of increased mitochondrial oxidation as the primary metabolic fate of epithelial pyruvate in human PCa, which has not been reported previously.

An alternative approach to testing the same hypothesis in a more clinically applicable way was to assess a list of differentially enriched metabolites by building a DESI-MSI metabolic classifier of benign and malignant epithelial regions. Using metabolite data from the same three KEGG pathways, we first noted a clear unsupervised separation between the benign and tumour epithelial cell clusters shown in Fig. 4d. We then trained and tested a deep-learning-based metabolic classifier (Fig. 4e; Code Availability) that achieved a median performance of 0.91 for differentiating between benign and tumour epithelial ROIs following five-fold cross-validation (Fig. 4f). Using Shapley

**Fig. 3 | The role of mitochondrial pyruvate import in PCa metabolic reprogramming and its impact on clinical [1-¹³C]lactate labelling. a** Schematic representation of the proposed differences in the metabolic fate of [1-¹³C]pyruvate between the benign and malignant prostatic epithelium, with the latter showing increased mitochondrial pyruvate uptake via AR-regulated MPCs to fuel both the restored TCA cycle and FASN-catalysed fatty acid biosynthesis. **b** Representative fluorescent RNAscope images of epithelial mRNA *LDHA* (gold) and *LDHB* (white) expression, along with IHC images of epithelial MPC2, FASN, and AR expression in the benign and malignant glands. **c** Scatterplots with bars comparing the expression of epithelial LDH, MPC2, FASN, and AR density between the benign ($n = 11$ samples) and tumour ($n = 11$ samples) areas from the HP-¹³C-MRI cohort ($n = 7$ patients). **d** Top: Plots comparing the log-transformed epithelial and stromal MPC1 and MPC2 densities in the benign ($n = 11$ samples) and tumour ($n = 11$ samples) areas from the HP-¹³C-MRI cohort ($n = 7$ patients). Bottom: Spearman's correlation plots comparing tumour epithelial MPC2 density against tumour epithelial FASN and AR

densities, as well as HP-¹³C-MRI-derived [1-¹³C]lactate SNR. **e** Mixed box-and-whisker and scatterplots comparing TCGA-PRAD derived bulk mRNA expression of *LDHA*, *LDHB*, total LDH, *MPC2*, *FASN*, and *AR* between benign ($n = 45$) and tumour ($n = 293$) prostatectomy samples from $n = 293$ patients. **f** Average expression dot plots comparing single-cell RNA-seq epithelial expression of the same genes from publicly available EGAS00001005787 (left) and GSE176031 (right) datasets. **g** H&E map of a surgical specimen including areas of benign and malignant prostate with corresponding spatial transcriptomics maps demonstrating the expression of total LDH, *MPC2*, and *FASN* obtained from a publicly available EGAS00001006124 dataset. In **c**–**e**, lines are median values and bars are interquartile ranges, with *P* derived using the two-sided Mann–Whitney U test or Wilcoxon signed-rank test, as appropriate. Scale bars denote 5–10 μm. Source data are provided as a Source Data file. **d** was created with BioRender.com and released under a Creative Commons Attribution-NonCommercial-NoDerivs 4.0 International license.

additive explanations[59], we identified the list of the ten most important metabolites for the classification, all presented in Fig. 4e. In keeping with the MPEA results, six of these were fatty acids, and these were more abundant in correctly classified tumour epithelial ROIs (see Supplementary Fig. 2a, showing consistently increased tumour palmitoleic acid abundance across all patients). The remaining four metabolites were glucose, malate, citrate, and malonate, and in comparison, were in high abundance within benign ROIs. Unsurprisingly, lactate had little impact on the developed classifier, ranking 28/31 in the list of the most important metabolites, with a mean absolute SHAP value of 0.009 (for comparison, the same value for oleic acid was 0.056). Overall, these results complement both the MPEA findings and tissue biomarker expression from the HP-¹³C-MRI cohort, suggesting that increased tumour mitochondrial pyruvate flux is an important metabolic feature of human PCa that can be probed both in vivo and ex vivo using metabolic imaging techniques. However, the impact of epithelial mitochondrial pyruvate import on hyperpolarised [1-¹³C] lactate labelling can only be assessed in tumour but not benign tissue, due to the lack of discernible HP-¹³C-MRI signal in the less cellular and vascular benign prostate.

### Cribriform PCa may show no [1-¹³C]lactate signal due to high MPC density and low LDH expression

Having explored the mechanisms behind the ability of clinical HP-¹³C-MRI to detect PCa by distinguishing it from the healthy tissue, we then assessed the differential [1-¹³C]lactate labelling between tumours harbouring cribriform and non-cribriform GP4 disease. As shown in Table 1, Fig. 5a–c, and Supplementary Fig. 1b, 2/3 ISUP grade group (GG) 3 ICC lesions from the HP-¹³C-MRI cohort had no detectable [1-¹³C]lactate signal or measurable $k_{PL}$ despite sufficient [1-¹³C]pyruvate delivery and restricted diffusion on ¹H-MRI ADC, indicating high cellularity. Notably, these lesions were found in patients who also harboured two contralateral small-volume ISUP GG2 non-cribriform tumours, which were both detectable on HP-¹³C-MRI (Fig. 5a, b and Supplementary Fig. 1b), therefore suggesting that the absence of [1-¹³C] lactate signal or measurable $k_{PL}$ in these large cribriform lesions was not due to technical issues.

To explain these findings within the same biological validation framework applied to the prior research questions above, we compared the metabolic and non-metabolic drivers of [1-¹³C]lactate labelling between the HP-¹³C-MRI-visible and HP-¹³C-MRI-occult tumours. Interestingly, while epithelial cell density, CD31-derived MVD, and epithelial MCT1 expression were similar between the two tumour types (Fig. 5d, e), ICC lesions with no discernible [1-¹³C]lactate labelling or measurable $k_{PL}$ showed almost negligible epithelial LDH density together with a marked increase in epithelial MPC2, FASN, and AR expression (Fig. 5d, e). These findings provide additional support for the hypothesis that tumour [1-¹³C]lactate labelling reflects the differential intracellular metabolic fate of pyruvate between LDH-catalysed

reduction to lactate, and MPC-driven mitochondrial import to facilitate oxidative metabolism. Moreover, this observation highlights the multifactorial nature of signal generation with HP-¹³C-MRI, including both metabolic and physical or physiological tissue characteristics. Specifically, if non-metabolic factors such as epithelial cell density and vascularity can be corrected for, then tumour [1-¹³C]lactate labelling patterns may reflect the true metabolic differences between lesions of different histological phenotypes.

### Gleason pattern 4 PCa consists of lipogenic cribriform and glycolytic non-cribriform metabolic phenotypes

Given the absence of HP-¹³C-MRI signal in some cribriform lesions in this cohort, we conducted a granular comparative metabolic analysis of different Gleason pattern glands included in this study. We previously demonstrated a significant positive correlation between tumour [1-¹³C]lactate labelling and the percentage of GP4 (%GP4) disease in the HP-¹³C-MRI cohort[2]. However, since all but one HP-¹³C-MRI-visible lesion consisted of non-cribriform GP4 glands, this reported relationship actually existed between [1-¹³C]lactate SNR and the percentage of non-cribriform GP4 disease. When all 15 lesions were divided using the %GP4 cut-off of 10%, no difference in [1-¹³C]lactate labelling was noted between high %GP4 and low %GP4 tumours ($P = 0.46$; Fig. 6a). Conversely, when the high %GP4 group was further subdivided by the dominant GP4 subtype, ICC or non-cribriform, the three resulting tumour types all showed significantly different [1-¹³C] lactate labelling ($P < 0.05$ for all; Fig. 6a) in the absence of notable changes in tumour ADC, $K^{trans}$, and epithelial MCT1 density ($P > 0.05$ for all; Fig. 6a).

We subsequently measured the expression of key metabolic biomarkers at the level of individual GP3 ($n = 50$), GP4 non-cribriform ($n = 22$), and GP4 ICC ($n = 17$) glands. Total epithelial LDH expression was highest in non-cribriform GP4 glands compared to both GP3 and GP4 ICC ($P < 0.0001$ for both; Fig. 6b, c), with no difference noted between the latter two morphological subtypes ($P = 0.4$; Fig. 6b, c), mirroring the trends noted in Fig. 6a for [1-¹³C]lactate labelling across histological types. At the transcriptional level, this observation was also in line with a similar pattern of nuclear HIF-1α expression, which was also significantly increased in non-cribriform GP4 glands compared to the other glands ($P < 0.05$ for both; Fig. 6b, c). Conversely, both MPC2 and FASN increased significantly in a stepwise fashion across the three Gleason phenotypes, with a similar trend noted for the epithelial nuclear AR expression ($P < 0.05$ for all; Fig. 6b, c). Overall, these results confirm that clinical [1-¹³C]lactate labelling reflects the tumour-specific pyruvate metabolic fate, which appears to favour a reduction to lactate in non-cribriform lesions and towards mitochondrial flux in cribriform GP4 glands.

These tissue-based metabolic metrics were then compared with endogenous metabolite measurements by dividing lesions from the DESI-MSI cohort into similar groups depending on the percentage and

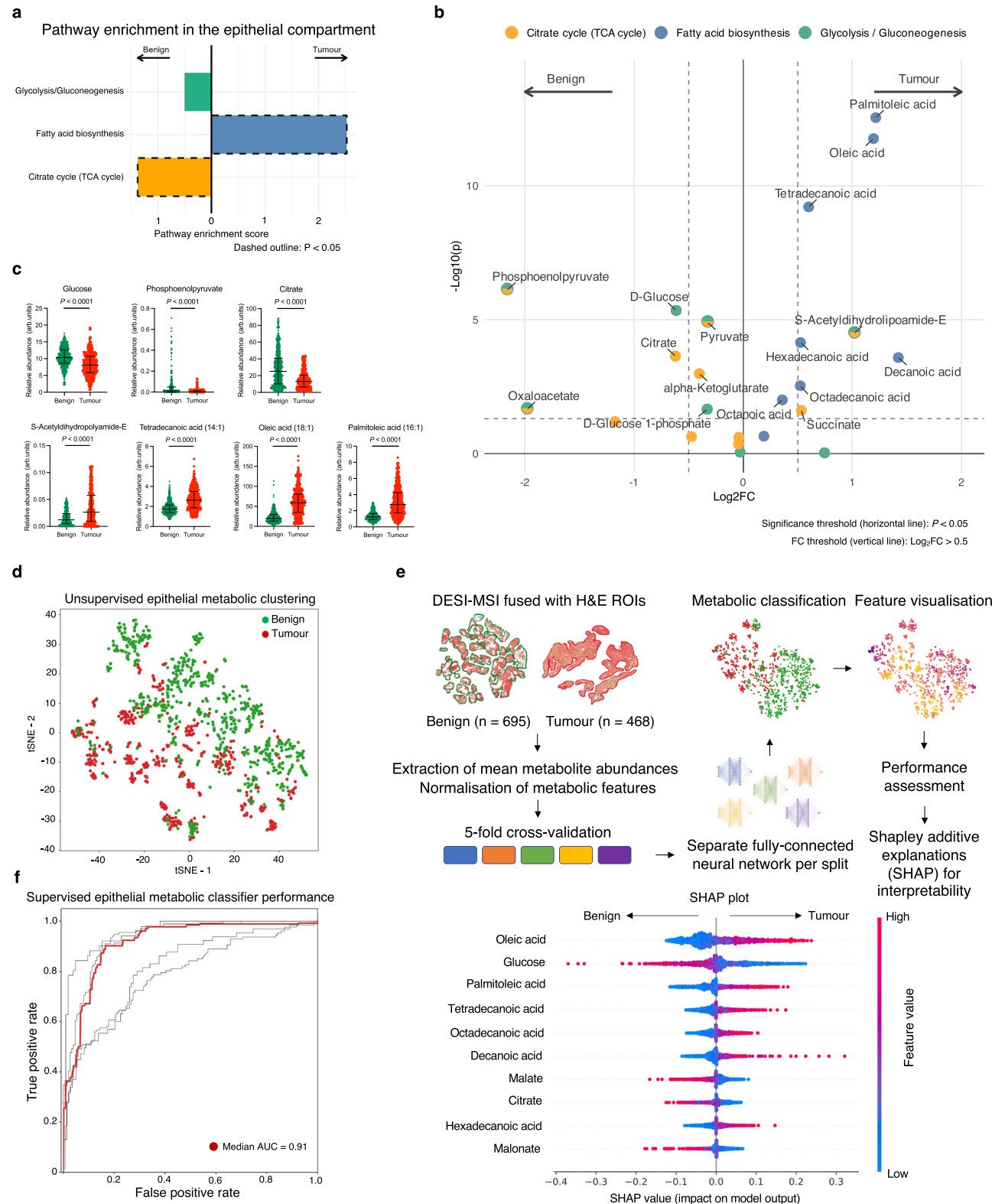

morphology of GP4 disease. In line with the [1-¹³C]lactate labelling pattern shown in Fig. 6a, DESI-MSI-derived epithelial lactate was similar between cores derived from low ($n = 19$) and high ($n = 19$) %GP4 lesions ($P = 0.93$; Fig. 7a). Moreover, when the cribriform and non-cribriform lesions were separated, the epithelial lactate pool was also found to be significantly increased in non-cribriform high %GP4 lesions compared to both high %GP4 ICC tumours and low %GP4 disease ($P < 0.05$ for all; Fig. 7a, b and Supplementary Fig. 2b where patient-to-patient variation

illustrates the differences between the three phenotypes). In contrast to the HP-¹³C-MRI cohort, high %GP4 ICC lesions had the lowest epithelial lactate abundance and showed decreased epithelial cellularity, which was significantly reduced compared to high %GP4 non-cribriform tumours and was similar to low %GP4 PCa ($P = 0.04$ and 0.26, respectively; Fig. 7a). Compared to cribriform lesions from the HP-¹³C-MRI cohort, which required a sufficient epithelial cell density to be ¹H-MRI-visible and therefore eligible for study inclusion[2], this is

**Fig. 4 | Spatially resolved metabolic profiling of the benign and malignant prostatic epithelium. a** Outputs of the DESI-MSI-derived MPEA demonstrating comparative enrichment of KEGG glycolysis, fatty acid biosynthesis, and TCA cycle pathways in benign ($n = 695$) and tumour ($n = 468$) ROIs from the spatial metabolomics cohort ($n = 13$ patients). **b** Volcano plot showing individual differentially enriched metabolites from the three KEGG pathways between the benign and tumour ROIs; $P$ were derived using the FDR-corrected Wilcoxon rank sum test. **c** Mixed box-and-whisker and scatterplots comparing epithelial abundance of the key differentially enriched metabolites between benign and tumour epithelial ROIs; the data are presented as median and interquartile range; $P$ were derived using the two-tailed Mann–Whitney $U$ test; the sample size in each plot varies depending on the number of outliers excluded using the ROUT method (Q = 5%). For illustrative purposes, outliers were removed using the ROUT method[104] with Q = 5%. **d** tSNE plot of DESI-MSI data acquired from the benign and tumour ROIs focusing on ions corresponding to metabolites related to the three KEGG pathways. **e** Summary diagram describing the key steps in developing a deep learning based metabolic tissue classifier using DESI-MSI derived metabolites from the three KEGG pathways to discriminate between benign and tumour epithelial ROIs; the SHAP plot lists the ten most important metabolites used by the final model to achieve a median performance of 0.91 presented in an AUC plot in **f**. Source data are provided as a Source Data file.

more representative of routine clinical practice where only 17% of pure cribriform lesions are detected on conventional imaging due to their loose cellular composition[18,60]. Hence, considering the importance of epithelial cell density in generating clinical HP-$^{13}$C-MRI signal, as well as the low epithelial LDH density in 2/3 cribriform lesions in the primary cohort, the low tissue lactate pool in ICC lesion highlights the potential limitation of using HP-$^{13}$C-MRI for detecting cribriform PCa and distinguishing it from less aggressive tumour subtypes.

Conversely, in line with the FASN expression trend, epithelial palmitoleic acid showed a similar stepwise increase across the three tumour phenotypes that was independent of epithelial cellularity ($P < 0.05$ for all; Fig. 7a, b and Supplementary Fig. 2b for patient-to-patient assessment). This observation aligned with the MPEA results (Fig. 7c), which compared the enrichment of KEGG glycolysis, TCA cycle, and fatty acid biosynthesis pathways across individual benign/GP3 glands ($n = 360$), non-cribriform GP4 glands ($n = 70$), and GP4 ICC glands ($n = 36$). Specifically, fatty acid biosynthesis was consistently enriched in more aggressive disease, and highest in GP4 ICC glands (Fig. 7c), in line with a recent study involving transcriptomic profiling of cribriform lesions[16]. Conversely, epithelial lactate as part of the glycolytic pathway was significantly enriched in non-cribriform GP4 tumours compared to both benign and GP3 glands, as well as GP4 ICC lesions ($P < 0.05$ for both; Fig. 7c). As summarised in Fig. 7d, these results indicate that GP4 PCa is comprised of two distinct metabolic phenotypes: lactate-rich, less aggressive non-cribriform tumours and lipid-rich, more aggressive ICC lesions.

## Discussion

HP-$^{13}$C-MRI is an emerging clinical tool to non-invasively image tissue metabolism. This exploratory clinical study deployed an array of in vivo and ex vivo metabolic imaging techniques to explore the biological mechanisms underpinning the ability of HP-$^{13}$C-MRI to study human PCa. This has demonstrated how metabolic signatures within PCa can vary between histological subtypes and has revealed two metabolically distinct phenotypes within intermediate-risk GP4 disease. Our results highlight several important considerations for future clinical translation of HP-$^{13}$C-MRI both in PCa and in other tumour types and could have important implications for characterising tumours in the future.

First, we show that clinical [1-$^{13}$C]lactate labelling offers a measure of absolute tissue lactate abundance, which can be influenced by both metabolic and non-metabolic (e.g., cell density and blood flow) tissue properties. For example, the increase in the absolute tissue lactate pool in HP-$^{13}$C-MRI-visible PCa compared to either occult PCa or benign prostate can be partly explained by the increased number and density of tumour epithelial cells. However, further factors play a role as two cribriform lesions were occult on imaging suggesting that epithelial cell density and vascularity are insufficient for generating clinical [1-$^{13}$C]lactate labelling in tumours in the context of low epithelial LDH expression, reduced endogenous lactate pool, and increased capacity for mitochondrial pyruvate uptake. These results highlight the importance of using a similar biological validation framework in future clinical HP-$^{13}$C-MRI studies to understand the exact

mechanisms underlying the observed changes in [1-$^{13}$C]lactate labelling in different cancers and various clinical scenarios. For example, since MPC2 is directly regulated by AR, early metabolic response to androgen deprivation therapy (ADT) could involve increased tumour [1-$^{13}$C]lactate labelling due to reduced [1-$^{13}$C]pyruvate mitochondrial uptake; we have previously shown a similar early increase following treatment in human breast cancer[9,61]. Conversely, a later reduction in tumour [1-$^{13}$C]lactate signal demonstrated in a case report by Aggarwal *et al.*[6] may reflect a combination of impaired tumour perfusion due to ADT-induced endothelial damage[62], and decreased tumour epithelial cell density[63] due to the loss of glandular architecture and apoptotic cell death[64]. Similarly, if HP-$^{13}$C-MRI is used for imaging response to poly (ADP-ribose) polymerases (PARP) inhibitors, early post-treatment scans may show increased [1-$^{13}$C] lactate labelling due to replenished NAD$^+$ levels, 90% of which is consumed by DNA-damage activated PARPs[9,65]. Importantly, the ability to resolve the mechanism of post-ADT [1-$^{13}$C]lactate labelling changes could be further improved by optimising the capacity for imaging hyperpolarised $^{13}$C-bicarbonate in the tumour. While in this study we did not reliably detect $^{13}$C-bicarbonate formation in the benign and malignant prostate, future work could maximise the detection of bicarbonate by truncating the imaging time to retain sufficient polarisation to allow for slice-localised magnetic resonance spectroscopy (MRS) following imaging, along with utilising a higher flip angle acquisition to increase the $^{13}$C-bicarbonate SNR. We are not aware of any published clinical study reporting the detection of $^{13}$C-bicarbonate in the human prostate, and therefore detecting it using MRS would provide evidence to optimise existing spiral or echo-planar spectroscopic imaging (EPSI) sequences in future studies. Important work in this area could also be facilitated by the recently published clinically translatable method of $^{13}$C-bicarbonate imaging that showed good performance in preclinical models of PCa[66]. Successfully translating these efforts into clinical studies will expand the clinical potential of HP-$^{13}$C-MRI considering the important role of altered mitochondrial pyruvate flux in human PCa development. In addition, increasing spatial resolution would also allow us to eliminate unnecessary variability in the [1-$^{13}$C]pyruvate signal derived from the vascular and extracellular compartments, thereby improving the quantification of $k_{PL}$ as a more reliable HP-$^{13}$C-MRI metric that is independent of polarisation levels.

The lack of [1-$^{13}$C]lactate signal in some aggressive cribriform lesions, which are known to harbour increased genomic instability[67], may also be explained by a decreased NAD$^+$ pool in addition to low LDH expression, and an increase in MPC-driven mitochondrial [1-$^{13}$C]pyruvate uptake. While non-invasive assessment of the percentage of lactate-rich non-cribriform GP4 glands may be helpful in some clinical settings such as active surveillance (AS), accurate detection and quantification of cribriform disease is of high diagnostic importance for several reasons. First, it has been shown that patients with non-cribriform ISUP GG2 disease have the same prognosis as men with ISUP GG1 lesions, whereas those with an ICC component had more frequent surgical and radiotherapy failure[68,69]. The European Association of Urology PCa guidelines recommend against offering AS

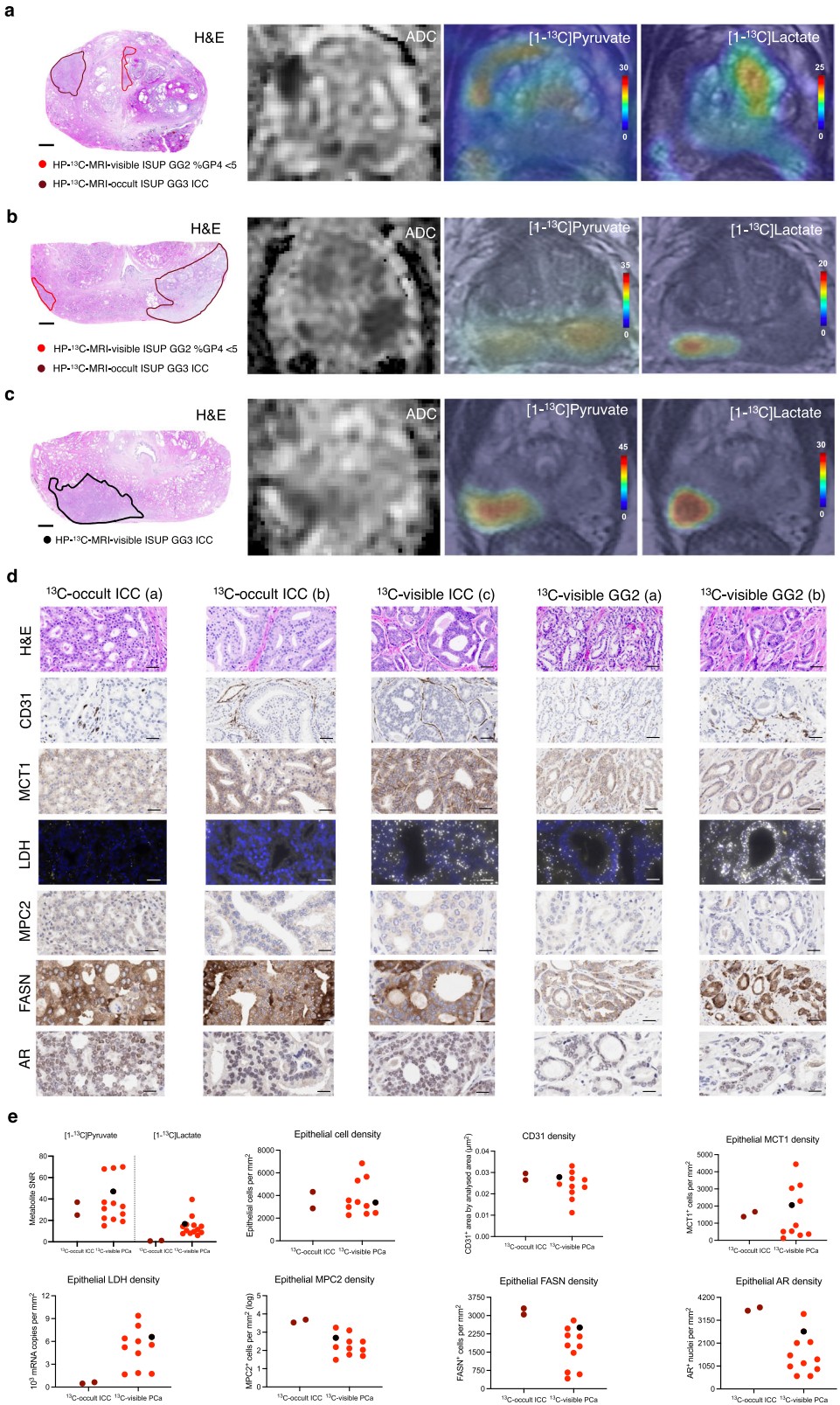

to patients with even small amounts of ICC detected on diagnostic biopsies[70,71]. The absence of [1-$^{13}$C]lactate signal in high-probability lesions as shown in Fig. 5a, b might raise the suspicion for the underlying cribriform morphology. However, as most cribriform lesions are difficult to visualise on standard $^{1}$H-MRI[18,72], the potential reduction in their [1-$^{13}$C]lactate labelling may limit the utility of HP-$^{13}$C-MRI for increasing their detection rate. Furthermore, in cases of mixed lesions

with other Gleason patterns present, as well as intraductal carcinoma (IDC) that is often composed of a cribriform component, lactate-poor ICC glands may artificially decrease tumour [1-$^{13}$C]lactate labelling with potential implications for reducing the accuracy of lesion characterisation. Conversely, the results here suggest that future development of novel $^{13}$C-labelled probes targeting fatty acid metabolism may yield important results given the continuous upregulation of lipid

**Fig. 5 | Comparative assessment of biological factors underpinning HP-¹³C-MRI-visibility of biopsy-proven PCa. a, b** Whole-mount H&E, ADC, along with [1-¹³C] pyruvate and [1-¹³C]lactate SNR maps demonstrating the presence of large, cellular ISUP GG3 lesions with dominant ICC component that were HP-¹³C-MRI occult compared to contralateral small-volume foci of ISUP GG2 disease with <5% non-cribriform GP4 glands. **c** Comparator case of a HP-¹³C-MRI-visible large cribriform ISUP GG3 tumour that was also visible on ¹H-MRI ADC. **d** Representative H&E slides, along with corresponding IHC-derived CD31, MCT1, MPC2, FASN, AR, and RNAscope-derived total LDH images obtained from HP-¹³C-MRI-visible and HP-¹³C-

MRI-occult lesions shown in **a–c. e** Mixed box-and-whisker and scatterplots comparing [1-¹³C]pyruvate and [1-¹³C]lactate SNR, epithelial cell density, CD31 density, as well as epithelial MCT1, LDH, MPC2, FASN, and AR density between HP-¹³C-MRI-visible (*n* = 13 samples) and HP-¹³C-MRI-occult (*n* = 2 samples) lesions from the hyperpolarised MRI cohort (*n* = 8 patients). **a–d** include images from three separate patients; imaging and staining were not repeated. In **e**, the data points for the ¹³C-visible ICC tumour are coloured in black. Scale bars in **a–c** and **d** denote 5 mm and 5–50 μm, respectively. Source data are provided as a Source Data file.

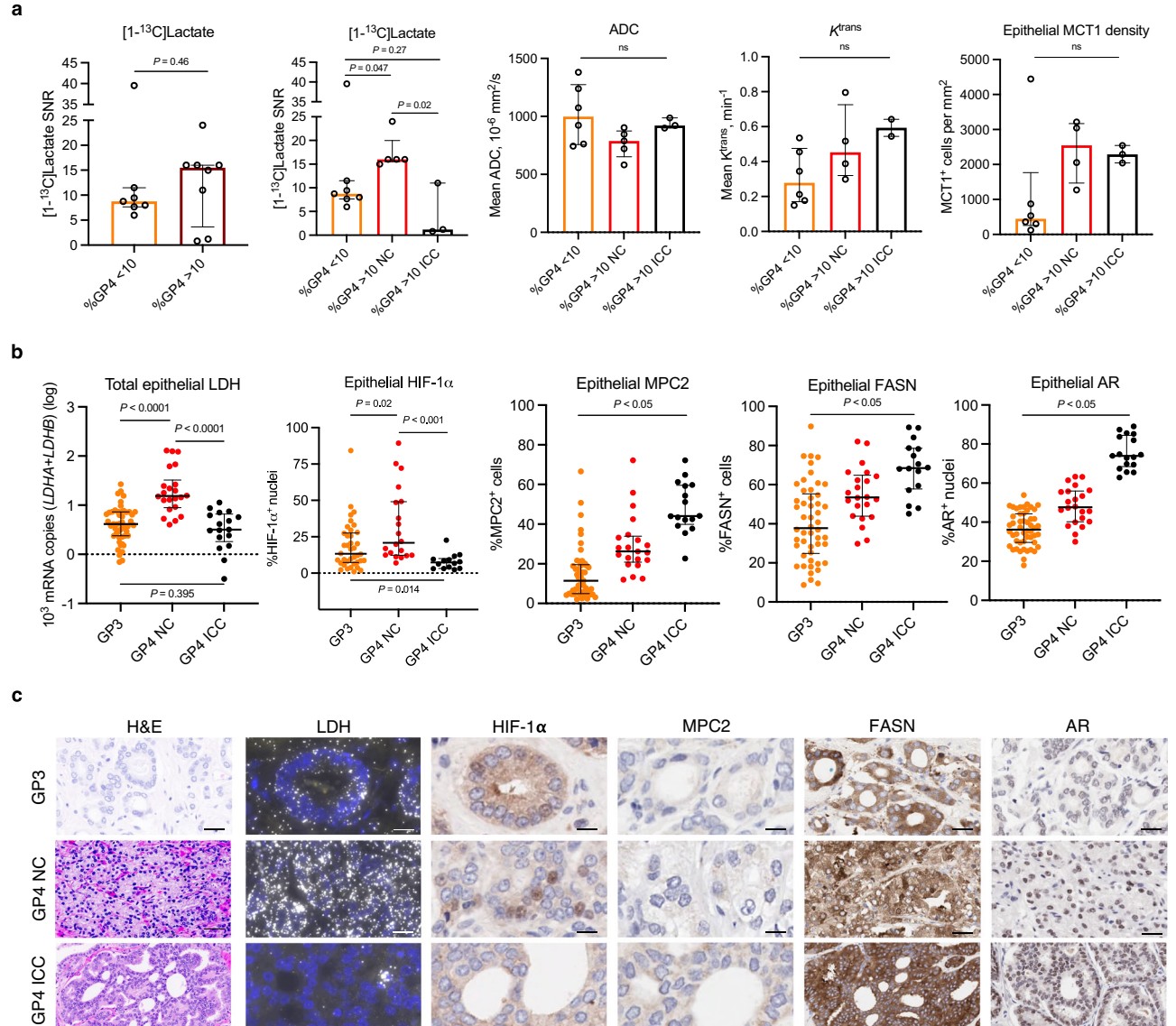

**Fig. 6 | Comparative metabolic characterisation of intermediate-risk PCa with varying percentage and phenotype of GP4 disease. a** Scatterplots with bars comparing clinical [1-¹³C]lactate labelling between tumours divided based on the percentage (far left; *n* = 7 samples for low %GP4 and *n* = 8 samples for high %GP4 lesions, respectively) and histological subtype (second from left; *n* = 7 samples for low %GP4, *n* = 5 samples for non-cribriform high %GP4, and *n* = 3 samples for cribriform high %GP4 lesions, respectively) of GP4 disease from *n* = 8 patients. Intergroup comparisons of ¹H-MRI-derived tumour ADC and *K*^trans, as well as tissue-based epithelial MCT1 density are also presented. **b** Mixed box-and-whisker and

scatterplots comparing tissue-based total epithelial LDH, nuclear HIF-1α, MPC2, FASN, and nuclear AR between ROIs harbouring individual GP3 (*n* = 50), non-cribriform GP4 (*n* = 22), and GP4 ICC (*n* = 17) glands from *n* = 8 patients.
**c** Representative H&E, RNAscope, and IHC images illustrating differential expression of tissue-based metabolic biomarkers between the three Gleason pattern glands. In **a, b**, the data are presented as median (denoted by the bars or boxes) with interquartile range (denoted by the bars or whiskers). *P* were derived using the two-sided Mann–Whitney *U* test. Source data are provided as a Source Data file. Scale bars denote 5–50 μm.

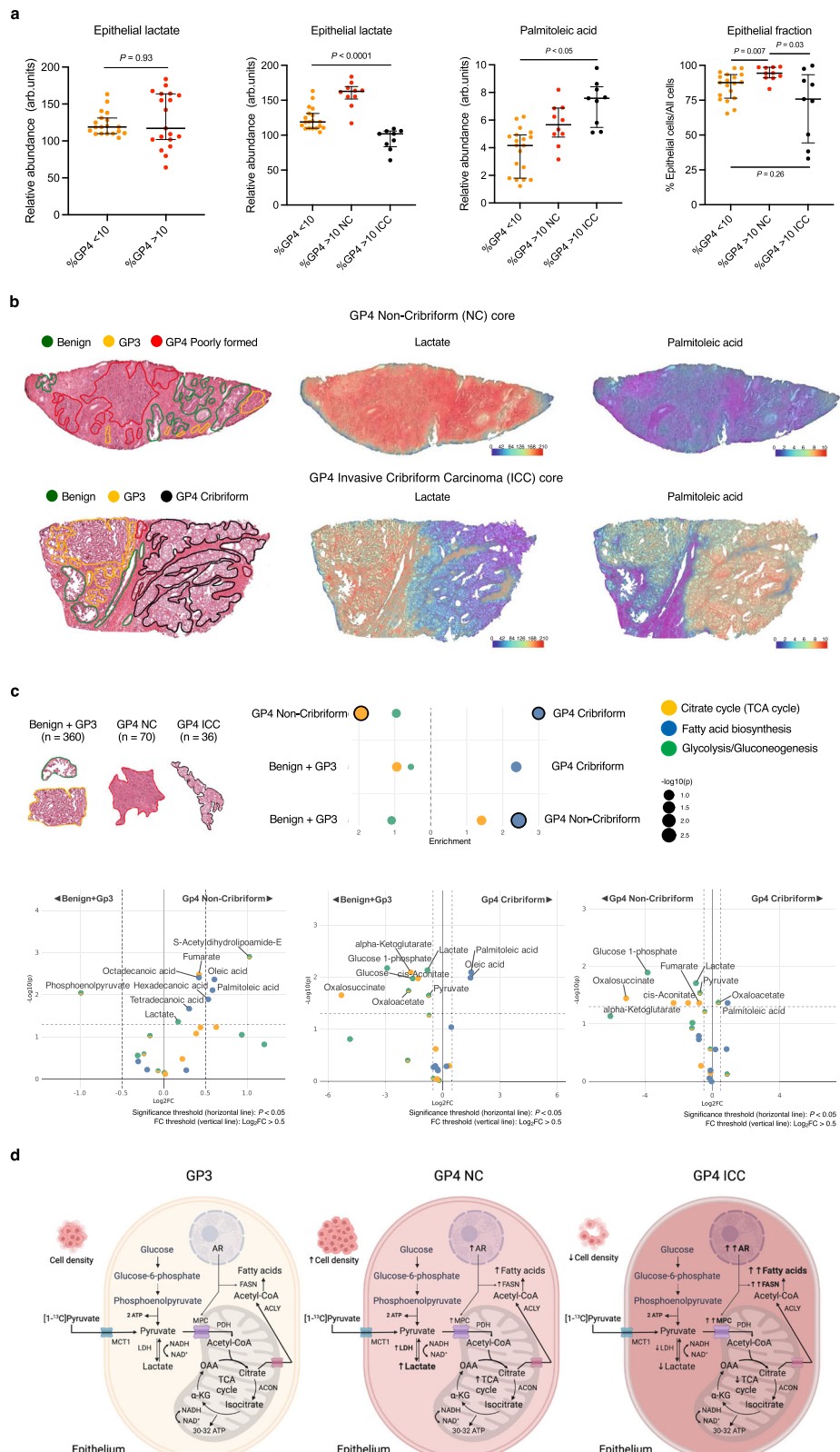

metabolism noted throughout PCa development[46,73] and the subsequent increase in endogenous fatty acid abundance compared to small metabolites such as lactate. Future studies could further assess the role of PET agents such as [11]C-acetate and [11]C-choline for detecting ICC and IDC, as well the applicability of prostate-specific membrane antigen (PSMA) PET for detecting cribriform PCa where there are currently conflicting results[74–76].

This study adds a new dimension to the rapidly growing body of literature investigating the molecular heterogeneity of different GP4 phenotypes by demonstrating the clinical potential for metabolic characterisation of intermediate-risk disease. Recent studies including bulk[16,17] and single-cell[15] mRNA sequencing have not focused on metabolic gene sets, and clinical studies involving spatial metabolomics techniques[35,77–80] have not distinguished between GP4 subtypes.

**Fig. 7 | Spatially resolved metabolic phenotyping of intermediate-risk human PCa. a** Mixed box-and-whisker plots comparing DESI-MSI derived epithelial lactate abundance between cores derived from lesions with low %GP4 (n = 19 samples) and high %GP4 (n = 19 samples) (far left), as well as between cores sub-stratified by the division of high %GP4 lesions into those with dominant non-cribriform (n = 10 samples) and cribriform (n = 9 samples) GP4 component (second left) from n = 13 patients. Plots demonstrating intergroup comparison of epithelial palmitoleic acid pool and epithelial cell fraction are also presented; the data are presented as median (boxes) with interquartile ranges (whiskers); P were derived using the two-sided Mann–Whitney U test. **b** Representative fused H&E and DESI-MSI lactate and palmitoleic acid images demonstrating differential metabolite abundance between GP3 (yellow), GP4 non-cribriform (red), and GP4 cribriform (black) glands. **c** MPEA outputs demonstrating differential enrichment of glycolysis, TCA cycle, and fatty acid biosynthesis KEGG pathways and individual metabolites between ROIs classified as benign and GP3 (n = 360), non-cribriform GP4 (n = 70), and cribriform GP4 (n = 36) glands from n = 13 patients; P were derived using the FDR-corrected Wilcoxon signed-rank test. **d** Diagram summarising the key results of this study and highlighting the presence of lactate-rich non-cribriform and lactate-poor, fatty acid-rich cribriform GP4 phenotypes within intermediate-risk PCa. Source data are provided as a Source Data file. **d** was created with BioRender.com and released under a Creative Commons Attribution-NonCommercial-NoDerivs 4.0 International license.

Our future work will build on the current results by identifying transcriptomic drivers behind the described metabolic reprogramming in different GP4 phenotypes, as well as GP3 and GP5 glands. If correlated with clinical outcomes and imaging findings, this work may yield important insights into metabolic characteristics of clinically significant PCa, thereby navigating the development of more specific metabolic imaging probes beyond [1-13C]pyruvate.

The limitations of this study include the lack of a direct correlation between HP-13C-MRI and spatial metabolomics data in the same cohort, which was due to the challenges in using formalin-fixed and paraffin-embedded tissue for DESI-MSI. The current study has compensated for this by careful histopathological matching between the two prospective patient cohorts, with the key results being consistent both with the two patient groups and with previously published data. While the HP-13C-MRI cohort had a limited sample size, this was in keeping with recent clinical reports by other centres that have the capacity to perform clinical HP-13C-MRI studies in patients with PCa[4,30,81]. In addition, this study did not include patients with GP5 disease or IDC as these rarely undergo surgical treatment, which will be assessed in future work. Importantly, future studies will include larger patient cohorts to assess the generalisability of our exploratory findings and can be statistically powered based on the results here. In addition, future studies can also investigate the role of other metabolic pathways in differentiating PCa subtypes. Moreover, while functional validation of the impact of MPC expression on mitochondrial pyruvate flux in PCa cells was outside of the scope of this clinical study, future work using stable isotope tracing in cell lines, organoids, or patient-derived xenografts (PDXs) of cribriform and non-cribriform disease would be key for confirming and explaining the mechanisms underlying the main findings outlined in this study[82]. While similar work has been conducted in cardiomyocytes[55], taking this forward in the PCa setting along with advanced lipidomic analysis could provide a more comprehensive understanding of the metabolic changes associated with mitochondrial pyruvate flux in terms of TCA reactivation compared to the lipogenic phenotype. Finally, while this study relied on whole-mount sections for three-dimensional matching between clinical HP-13C-MRI and tissue-based digital pathology, IHC, and spatial transcriptomics, future work in non-surgical patient populations should consider alternative approaches (e.g. fiducial marker placement) to ensure that targeted biopsy samples truly represent the tissue biology underlying HP-13C-MRI annotations.

In conclusion, this study demonstrates the impact of metabolic and non-metabolic tissue characteristics on the characterisation of PCa using HP-13C-MRI. Using a combination of macroscopic and microscopic imaging techniques, we show that clinical [1-13C]lactate labelling is a function of epithelial cell density, vascularity, cellular capacity for pyruvate reduction to lactate, and MPC-driven mitochondrial pyruvate import. We have demonstrated that these characteristics differ significantly between both benign and malignant prostate tissue, and between intermediate-risk tumours containing cribriform and non-cribriform GP4 glands. If validated in larger

cohorts and expanded to other tumour subtypes, these results have important implications for the development and clinical translation of PCa metabolic imaging techniques.

## Methods
### Patient selection and ethics
This clinical study complies with all relevant ethical regulations and is approved by the National Research Ethics Service Committee East of England, Cambridge South, with the relevant study protocol numbers listed below. First, we analysed prospectively collected clinical, imaging, and tissue-based data obtained from patients from two separate surgical cohorts, as shown in Fig.1. The HP-13C-MRI cohort was a subset of a previously reported[2] population of consecutive male patients with at least one MR-visible (>1 cm, Likert/PI-RADS 4–5) histologically proven PCa focus (ISUP GG ≥ 2) scheduled for RARP in our centre. All patients provided written consent to participate in the original prospective study (Molecular Imaging and Spectroscopy with Stable Isotopes in Oncology and Neurology—Imaging metabolism in prostate [MISSION-Prostate] protocol), retrospective analysis of which was approved by the institutional review board (National Research Ethics Service Committee East of England, Cambridge South, Research Ethics Committee number 16/EE/0205). Of the ten patients recruited originally, eight were included in the current analysis due to technical failure of HP-13C-MRI in the remaining two patients[2]. Following a review of post-surgical whole-mount histological specimens, 15 lesions were reported, with their key morphological characteristics reported in Table 1.

To validate the key findings obtained in the HP-13C-MRI cohort by directly measuring tissue metabolite concentrations in fresh-frozen prostatectomy samples, we then retrospectively identified a secondary cohort of treatment-naïve surgical patients who had been originally recruited into an ethically-approved prospective national study (DIA-MOND, National Research Ethics Service Committee East of England, Cambridge South, Research Ethics Committee number 03/018) following the provision of written consent. To ensure comparability with the HP-13C-MRI cohort, a total of 13 patients with 15 lesions matching the key histopathological characteristics of tumours from the primary cohort (Table 1) were selected for analysis as part of this study. Further description of tumour histopathological assessment and matching is provided in the corresponding section below. All patients in both the MISSION-Prostate and DAMOND cohorts had male sex noted in their clinical records.

### HP-13C-MRI acquisition and analysis
In the primary cohort, pre-surgical HP-13C-MRI was acquired on a clinical 3.0 T MR system (MR750, GE Healthcare, Waukesha WI, USA) using a bespoke 1H/13C endorectal receive coil[83] following the protocol detailed in the original MISSION-Prostate report[2]. In brief, samples containing 1.47 g of [1-13C]pyruvic acid (Sigma Aldrich, St Louis MO, USA) and 15 mM electron paramagnetic agent (EPA) were hyperpolarised using a clinical hyperpolariser (SPINlab; 5 T Research Circle Technology, GE Healthcare, Waukesha WI, USA) by microwave

irradiation at 139 GHz at ~0.8 K for ~3 h followed by rapid dissolution in 38 mL of superheated sterile water and filtration to remove EPA to a concentration below ≤3 μM[2,8]. Radiofrequency pulses with a nominal flip angle of 15° were applied using a clamshell RF coil (GE Healthcare, Waukesha WI, USA) to acquire a 20 × 20 cm² FOV with a matrix size of 32 × 32 and a temporal resolution of 4 s for 20 time points. Images had a true in-plane resolution of 12.5 × 12.5 mm² and were reconstructed with a resolution of 128 × 128. The imaging data were reconstructed in MATLAB (MathWorks, Natick MA, USA)[2,8], with [1-¹³C]pyruvate and [1-¹³C]lactate SNR derived from ROIs containing areas of histopathologically-proven PCa ($n = 15$) and contralateral benign ($n = 15$) tissues. The ROIs were originally drawn on H&E whole-mount pathology (WMP) maps by an experienced genitourinary pathologist (A.Y.W.) (Supplementary Fig. 3). Corresponding lesions were then outlined on HP-¹³C-MRI metabolite maps (Supplementary Fig. 3) in consensus by two consultant/attending radiologists (T.B., F.A.G.) and a research fellow (N.S.) with 13-, 14-, and 5-years' experience in prostate MRI, respectively[84]. In addition, we calculated the apparent reaction rate constant for the exchange of the HP ¹³C-label between pyruvate and lactate ($k_{PL}$) using a two-site exchange model using a frequency-domain approach and linear least-squares fitting[2]. $k_{PL}$ quantification was important given the independence of this parameter from the polarisation level, which in this cohort varied between 7–33.8%, as indicated previously[2].

## ¹H-MRI acquisition and analysis

Either before or immediately after the HP-¹³C-MRI acquisition, all patients in the primary cohort underwent standard-of-care multiparametric MRI of the prostate. The protocol, detailed in the prior cohort description[2], included axial T₁-weighted fast spin echo (FSE), high-resolution T₂-weighted 2D fast recovery fast spin echo (FRFSE, echo time [TE] 98–107 ms, field-of-view [FOV] 22 × 22 cm², acquisition matrix 320–384 × 256, slice thickness 3 mm with 0 mm gap, 3 signal averages, repetition time [TR] 3000–5000 ms, echo train length 16, receiver bandwidth ± 31.25 or ± 41.67 kHz), diffusion-weighted imaging (DWI, spin-echo echo-planar imaging pulse sequence with b values of 150, 550, 750, 1000, and 1400, with a separate high b value acquisition of 2000 s/mm²), and dynamic contrast-enhanced MRI (DCE-MRI, axial 3D fast spoiled gradient echo [FSPGR], TR/TE 4.1/1.8 ms, FOV 24 × 24 cm², following bolus injection of gadobutrol [Gadovist, Bayer Healthcare, Berlin, Germany] via a power injector, rate 3 mL/s [dose 0.1 mmol/kg], temporal resolution 7 s). ADC maps were calculated automatically, representing the quantitative assessment of tissue density.

The mean ADC values corresponding to the areas of benign ($n = 15$) and malignant ($n = 15$) prostate were extracted from the same ROIs used for the previously described HP-¹³C-MRI analysis (Supplementary Fig. 5). The same ROIs were also used for extracting $K^{trans}$ values as measures of vascular perfusion and permeability (Supplementary Fig. 3), with DCE-MRI analysis performed in an in-house-developed MATLAB software that was used to generate B₁ maps[2,8]. These maps were then transferred to MIStar (Apollo Medical Imaging, Melbourne, Australia) to generate B₁-corrected T₁ maps, to perform motion correction of the DCE-MRI data using a 3D affine model, and for pharmacokinetic modelling using the standard Tofts model[2,8]. In patient 6 (Table 1), DCE-MRI was not performed for technical reasons.

## Histopathological assessment and digital pathology analysis

In the HP-¹³C-MRI cohort, prospectively obtained whole-mount surgical sections were formalin fixed and paraffin-embedded following the routine clinical procedure. The resulting FFPE slides were stained with H&E and digitised for subsequent histopathological assessment at 40x using Aperio CS2 Digital Pathology Scanner (Aperio Tech., Oxford, UK)[85]. The latter was performed by a fellowship-trained genitourinary pathologist who was a member of the 2019 ISUP Consensus Working Group on PCa grading[14] and has more than 20 years' experience of clinical PCa histopathological assessment. First, whole-mount slides were assessed for the presence of PCa foci and assigning their ISUP GG, quantifying %GP4, and identifying the dominant GP4 subtype in ISUP GG 2–3 lesions. As described in the previous sections, tumour ($n = 15$) and contralateral benign tissue ROIs ($n = 15$) were used for HP-¹³C-MRI and ¹H-MRI segmentation (Supplementary Fig. 5). The same ROIs were subsequently transposed onto IHC and RNAscope whole-mount slides for analyses described in the corresponding sections below. In addition, as described in the prior publication[2], whole-mount H&E maps were used to draw standardised[8] random ROIs encompassing tumour foci represented by clear GP3 ($n = 50$), non-cribriform GP4 ($n = 22$; poorly formed glands, fused glands, glomeruloid glands), and cribriform GP4 ($n = 17$; invasive cribriform carcinoma) glands (Supplementary Fig. 6); the identification of the latter was performed according to the 2021 ISUP consensus definition of cribriform pattern PCa[86]. These ROIs were also transposed onto IHC and RNAscope slides for the corresponding analyses. Importantly, to ensure adequate matching of ROIs drawn on the whole-mount H&E samples these were matched to the corresponding benign and malignant areas noted on HP-¹³C-MRI: whole-mount H&E slides were co-registered with ¹H-MRI-derived anatomical T₂-weighted images according to previously described methodology[87]. Specifically, the two types of images were matched by four readers (three urological imagers, N.S., T.B., and F.A.G., and one genitourinary pathologist, A.Y.W.) using anatomical landmarks, such as prostatic zonal boundaries, urethra, and pseudocapsules of benign hyperplastic nodules if present, while correcting for the distorting effects of specimen preparation where possible, as shown in Supplementary Fig. 7. This registration, as well as the whole-mount nature of surgical H&E samples and whole-organ MR images, enabled us to capture heterogeneity of both tumour and benign areas including in this analysis.

The key histopathological characteristics of tumours from the HP-¹³C-MRI cohort (ISUP GG, %GP4, dominant GP4 subtype) were used to guide the search for a matching set of 15 lesions to form the secondary DESI-MSI cohort. The original fresh-frozen surgical sample collection and preparation was performed by the same expert genitourinary pathologist (A.Y.W.) according to the previously described protocol[88]. In brief, RARP specimens were transferred on ice to the laboratory within 30 min of surgical resection, with multiple punch biopsies removed using a standard 4–6 mm skin punch. The sites of the punched cores were marked on a "map" diagram (Fig. 1), from which cores including benign ($n = 61$) and tumour tissues ($n = 56$) were selected for this study. H&E staining was performed following the FF sample preparation described in the DESI-MSI section below, with the resulting slides scanned at 40x using the same Aperio Digital Pathology Scanner. To ensure adequate preservation of tumour heterogeneity, all available tumour punch biopsies were included in this study, as shown in Fig. 1.

As in the HP-¹³C-MRI cohort, tissue assessment and annotation were performed in QuPath 0.23[89] by the same genitourinary pathologist (A.Y.W.). First, depending on the presence and amount of tumour tissue, each core was labelled as either benign (no tumour tissue, $n = 61$), mixed (<50% of tumour tissue, $n = 23$), and tumour (≥50% of tumour tissue, $n = 38$) (Supplementary Fig. 8) to ensure accurate comparison between benign and malignant prostatic tissues during the whole-core lactate abundance assessment detailed below. Second, the epithelial component of each individual core was annotated manually as shown in Fig. 2e and labelled according to its morphological characteristics (benign, GP3, GP4 non-cribriform, and GP4 ICC). This enabled us to assign the overall ISUP GG and %GP4, as well as identify the dominant GP4 subtype both at the level of an individual core and for the combination of cores obtained from a single large tumour focus. Both whole-core ($n = 69$) and individual epithelial ROIs

($n$ = 695 benign and $n$ = 468 tumour) were used to extract the DESI-MSI data as described in the relevant section below.

In both cohorts, spatially resolved tissue analysis at the level of epithelial and stromal compartments was critical due to the known metabolic compartmentalisation between the two tissue types in the human prostate. In the HP-$^{13}$C-MRI cohort, we used a previously described random forest tissue classifier[2] embedded into the HALO v3.2.1851.266 (Indica Labs, Albuquerque, NM, USA) software. In short, the classifier was trained using manual annotations of epithelial and stromal areas (Supplementary Fig. 9) according to the previously described methodological pipeline[51]. In the DESI-MSI cohort, we used the HALO AI platform to train a similar deep neural network (DenseNet V2, path size: 100 × 100 pixels) tissue classifier to again distinguish between epithelial and stromal cellular compartments. Since the DESI-MSI data was extracted from manually drawn epithelial ROIs, here the classification was only required for quantifying the relative distribution of epithelial and stromal compartments within a given core (quantified as the area of epithelial/stromal cells per the total tissue area, both in mm$^2$). To train the classifier, we used a subset of pathologist-defined annotations (Supplementary Fig. 6) of epithelial, stromal, and background regions from $n$ = 133 separate tissue regions with a total area of 171.4 mm$^2$. Classification outputs were visualised as mark-up images (Supplementary Fig. 6) and sent back to the expert pathologist (A.Y.W.) for corrections as part of an active learning process. The classifier performance was cross-validated against ground truth annotations using the unseen dataset of the validation cohort.

## Immunohistochemistry

To assess biological factors such as epithelial capacity for [1-$^{13}$C]pyruvate uptake, tissue vascularity, and inferred [1-$^{13}$C]pyruvate metabolic fate, we used IHC to stain the HP-$^{13}$C-MRI cohort samples for the following targets: MCT1 (membrane pyruvate importer; Cat. No. HPA003324, Atlas Antibodies, Bromma, Sweden), MPC1 and MPC2 (mitochondrial pyruvate carriers; Cat. No. PAB28306, Abnova, Taipei, Taiwan for MPC1; Cat. No. D417G, Cell Signalling Technology, Danvers MA, USA for MPC2), FASN (cytosolic fatty acid synthase; Cat. No. 3180, Cell Signalling Technology, Danvers MA, USA), AR (nuclear receptor regulating the expression of MPCs and FASN; Cat. No. NCL-AR-318, Novocastra, Newcastle, UK), HIF-1α (nuclear transcription factor regulating the expression of *LDHA*; Cat. No. ab51608, Abcam, Cambridge, UK), and CD31 (endothelial biomarker; Cat. No. M0823, Dako, Santa Clara CA, USA). All antibodies have been previously validated in our centre using positive and negative tissue controls under the supervision of specialist pathologists, as described in the Reporting Summary.

The staining was performed on FFPE prostatectomy tumour blocks using Leica's Polymer Refine Detection System (DS9800) in combination with their Bond automated system (Leica Biosystems Newcastle Ltd, Newcastle, UK) following a previously published protocol[2]. Briefly, sections were cut to 4 μm thickness and baked for 1 h at 60 °C ahead of deparaffinisation and rehydration, as standard, on the ST5020 Multistainer (Leica Biosystems). Subsequent immunohistochemical staining was carried out on Leica's automated Bond III platform (Leica Biosystems) in conjunction with their Polymer Refine Detection System (Cat. No. DS9800, Leica Biosystems). Sections stained for HIF-1α were pre-treated with Epitope Retrieval Solution 1 (Cat. No. AR9961, Leica Biosystems) and those stained for other antibodies were pre-treated with Epitope Retrieval Solution 2 (Cat. No. AR9640, Leica Biosystems). Incubation was for 20 min at 99 °C. Antibodies were diluted to 23.36 μg/mL (MCT1), 0.3 μg/mL (MPC1), 4.5 μg/ mL (MPC2), 1:100 (FASN), 1:50 (AR), 0.6 μg/ml (HIF-1α), and 4.1 μg/ml (CD31), respectively. Endogenous peroxidase activity was quenched using 3–4% (v/v) hydrogen peroxide and primary antibody was detected using Anti-rabbit Poly-HRP-IgG (<25 μg/mL; part of Leica Biosystems Polymer Refine Detection System) containing 10% (v/v)

animal serum in tris-buffered saline/0.09% ProClin 950. The complex was visualised using 66 mM 3,3'-Diaminobenzidine tetrahydrochloride hydrate in a stabiliser solution and ≤0.1% (v/v) Hydrogen Peroxide. DAB Enhancer (Cat. No. AR9432, Leica Biosystems) was used to intensify the signal. Cell nuclei were counterstained with <0.1% haematoxylin.

As described previously[2], HALO v3.3.2541.405 (Indica Labs, Albuquerque, NM, USA) Membrane v1.7 (MCT1), multiplex IHC v2.3.4 (MPC1, MPC2, HIF-1α), area quantification v2.2.1 (CD31), multiplex IHC v3.1.4 (AR), and multiplex IHC v2.3.4 (FASN) modules were used for automated analysis of scanned sections. Optical densities for weak, moderate, and strong stains were: MCT1, 0.1602, 0.2302, 0.4037; HIF-1α nuclear 0.1958, 0.7522, 0.885; CD31, 0.2164, 0.2721, 0.3832; AR nuclear, 0.0987,0.3761,0.544; FASN, 0.1881, 0.5949, 0.8407; MPC1, positive optical density threshold 0.0759; MPC2, positive optical density threshold 0.0758. Given the prior identification of the epithelial compartment as the source of clinical [1-$^{13}$C]lactate labelling in the HP-$^{13}$C-MRI cohort[2], we first quantified epithelial cell density of each of the above proteins by dividing the number of positive epithelial cells by the ROI areas measured in mm$^2$. This approach helped us achieve a closer comparison between the limited spatial resolution of absolute [1-$^{13}$C]lactate signal quantification on HP-$^{13}$C-MRI and the density of cells that could contribute to the observed signal. In addition, we also assessed the percentage of positive epithelial cells (MPC2, FASN) and nuclei (HIF-1α, AR) in individual epithelial glands harbouring GP3, non-cribriform GP4, and GP4 ICC disease to obtain a more refined metric of the biomarker expression at a cellular level.

All IHC antibodies used in this study were previously validated in our Histopathology Core Facility as described in the Reporting Summary. Here, in addition to routinely used MCT1, HIF-1α, FASN, and AR antibodies, we additionally validated antibodies for MPC1 and MPC2 within a dedicated Histopathology Core Facility (led by J.L.M.) under the supervision of an expert genitourinary pathologist (A.Y.W.), with further details and representative images from control tissues provided in Supplementary Fig. 10.

## Spatial transcriptomics

To infer epithelial capacity for LDH-catalysed [1-$^{13}$C]pyruvate-to-[1-$^{13}$C] lactate conversion, we used RNAscope to derive the total epithelial LDH density, measured as the total number of epithelial *LDHA* and *LDHB* mRNA copies per mm$^2$. The analysis was performed using the RNAscope spatial transcriptomics technique according to the previously described protocol[2]. Briefly, FFPE sections were cut to 4 μm thickness and baked for 1 h at 60 °C before loading onto a Bond RX instrument (Leica Biosystems Newcastle Ltd, Newcastle, UK). Slides were deparaffinized and rehydrated on board prior to pre-treatments using Epitope Retrieval Solution 2 (Cat No. AR9640, Leica Biosystems) at 95 °C for 15 min, and ACD Enzyme from the Multiplex Reagent kit at 40 °C for 15 min. Probes (*LDHA*, *LDHB*) were visualised using Opal fluorophores diluted to 1:1000 using RNAscope LS Multiplex TSA Buffer. Probe hybridisation, signal amplification, and detection were all performed on the Bond Rx according to the ACD protocol. Slides were removed from the Bond Rx and mounted using Prolong Diamond (Cat. No. P36965, ThermoFisher Scientific, Waltham, MA, USA). In all patients, simultaneous detection of human *LDHA* and *LDHB* was performed using Advanced Cell Diagnostics (ACD, Bio-Techne, Abingdon, UK) RNAscope 2.5 LS Multiplex Reagent Kit (Cat No. 322800), and RNAscope 2.5 LS probes (ACD, Hayward, CA, USA) validated by the manufacturer according to previously described procedures[90]. In addition, prior to the analysis, we used spare tissue sections to run the negative control slides (4 Plex DapB to ensure that DapB is in every channel) to assess background staining, along with the positive control slides (*POLR2A* for channel 1 and *PPIB* for channel 2) to determine good RNA quality. In the analysis optimisation, we used the negative controls to set the thresholds for positive signal in the test slides. Example

images are presented in Supplementary Fig. 11. The described routine in-house RNAscope antibody validation process, along with the subsequent analysis, was performed by an experienced member of our dedicated Histopathology Core Facility (see Acknowledgments) with 11-years' of experience of using all RNAscope automated kits available for the Leica Bond Rx (Single Plex, Duplex, 3 Plex, 4 Plex, BaseScope, and RNAscope Plus), as well as manual HiPlex kits, for more than 50 separate projects in a variety of tissues and species, including human and murine breast, brain, kidney, lung, and liver. The slides were imaged on the AxioScan (Carl-Zeiss-Stiftung, Stuttgart, Germany) to create whole-slide images. Images were captured at 40x magnification, with a resolution of 0.25 microns per pixel. HALO v3.2.1851.266 and the FISH v2.2.0 modules were used for the automated analysis of scanned RNAscope sections by an experienced member of our Core Facility (C.B.), which included specific steps aimed at overcoming autofluoresence as previously described[51].

### Spatial metabolomics acquisition and analysis

Fresh-frozen RARP specimens obtained from the DESI-MSI cohort were transferred on ice to the laboratory within 30 min of surgical resection, with multiple punch biopsies collected as described above. For the DESI-MSI experiment, the punched cores ($n = 117$) were embedded and prepared according to a previously reported sample preparation workflow[91]. In brief, the punched cores were co-embedded in a (Hydroxypropyl)-methylcellulose (HPMC) and Polyvinylpyrrolidone (PVP) hydrogel to enable time-efficient sectioning under comparable conditions for all specimens analysed in one experiment. A total of 18 punched cores were placed upright in peel-away moulds (Thermo Scientific, Waltham, MA, USA) pre-filled with ice-cold embedding medium. Snap freezing of the filled mould was performed in dry ice-chilled isopropanol followed by a wash in dry ice chilled iso-pentane to wash off the excess of isopropanol. The frozen moulds were kept on dry ice to allow evaporation of the adherent iso-pentane before sectioning. The resulting blocks ($n = 10$) were sectioned to 10 μm thickness using a CM3050 cryo-microtome (Leica Biosystems, Nussloch, Germany) and thaw-mounted onto Superfrost slides (Fisher Scientific, Loughborough, UK) for DESI-MSI and H&E histological examination.

DESI-MSI was carried out using an automated 2D DESI source (Prosolia Inc, Indianapolis, IN, USA) with home-built sprayer assembly mounted to a Q-Exactive FTMS instrument (Thermo Scientific, Bremen, Germany). Analyses were performed at spatial resolutions of 65 μm in negative ion mode and mass spectra were collected in the mass range of 80–600 Da with mass resolving power set to 70000 at $m/z$ 200 and an S-Lens setting of 100. Methanol/water (95:5 v/v) was used as the electrospray solvent at a flow rate of 1.0 μL/min and a spray voltage of −4.5 kV. Distance between DESI sprayer to MS inlet was 7 mm, while distance between sprayer tip to sample surface was 1.5 mm at an angle of 75°. Nitrogen N4.8 was used as nebulising gas at a pressure of 6.5 bar. Omnispray 2D (Prosolia, Indianapolis, USA) and Xcalibur (Thermo Fisher Scientific) software were used for MS data acquisition. Individual line scans were converted into centroided. mzML format using MSConvert (ProteoWizard toolbox version 3.0.4043) and subsequently into.imzML using imzML converter v1.3.

Raw metabolite spectra were extracted using the pyimzML python package and aligned to a common mass axis using a fixed bin size of 0.01 Da. Spectra were normalised by the root mean square of the intensity across the entire mass range, to compensate for signal instabilities and to facilitate comparison between experiments. The analysis of metabolites within pathologist-annotated H&E ROIs required the co-registration of whole-slide histological images with DESI-MSI data. To do so, we first extracted a low-resolution version of each whole-slide H&E-stained image, downsampling it by a factor of 32. A binary tissue-background mask was computed for each H&E

slide by thresholding the saturation of the image using Otsu thresholding. We then removed all objects and holes in the binary mask smaller than 64 pixels. For the DESI-MSI data, we computed a total ion current (TIC) image, normalised to lie in the range 0–1. The H&E binary mask was then resized to have the same dimensions as the MSI TIC image. Using the resized H&E mask and the MSI TIC image, we computed an affine transform using the Elastix package, treating H&E mask as the moving image and the MSI image as the fixed image. Pathologist annotations were converted to binary masks, resized to the same dimensions as the MSI total ion current image, and then transformed using the previously calculated affine transformation. This yielded a mask identifying the pixels in the MSI image corresponding to the region annotated by the pathologist on the whole slide H&E image. We then computed a single spectrum per ROI by taking the mean of all MSI pixels inside the region. Importantly, DESI-MSI was chosen over other spatial metabolomics techniques, such as matrix-assisted laser desorption ionisation (MALDI) MSI, as it rapidly acquires data and is particularly sensitive for the detection of small molecules which is the focus of this study[92]. However, to validate our DESI-MSI findings against an orthogonal technique, we performed MALDI-MSI in negative ion mode on serial sections of samples representing benign and malignant prostatic cores, which showed comparable mass accuracy for metabolite detection (Supplementary Table 2) and the key metabolites of interest showed a similar spatial localisation (Supplementary Fig. 12). In addition, we used tandem mass spectrometry (MS/MS)[93] to further validate the DESI-MSI derived targeted metabolite detection against metabolite standards. As shown in Supplementary Fig. 13, similar characteristic metabolite fragment peaks were observed in both pure control standards and human prostate test samples used in this study.

To resolve the metabolic phenotype of benign and tumour prostatic epithelial cells within the framework of this study, we used individual benign ($n = 695$) and tumour ($n = 468$) epithelial ROIs to derive relative abundance values of endogenous metabolites contributing to the three key metabolic pathways of interest (glycolysis, TCA cycle, fatty acid biosynthesis; individual KEGG metabolites are listed in Supplementary Table 1). The relative abundance values of all metabolites that have been mapped to the reference m/z values (within 5 ppm accuracy) in the KEGG database were compared between the groups of interest. The corresponding $P$ values have been calculated for each metabolite using non-parametric Wilcoxon rank sum test. To perform the metabolic pathway enrichment analysis (MPEA), the metabolites enriched ($P > 0.05$; Log2FC > 1) in either of the comparison groups were subjected to overrepresentation analysis. Only the pathways with an overall size of more than two metabolites and having more than two enriched metabolites contributing to the named pathway (or "hits") were retained. The produced output of MPEA contains a $P$ value based on a hypergeometric test, as well as the pathway enrichment score (PES) for each pathway. PES represents a log2 transformed ratio between the number of observed enriched metabolites in the dataset and the number of metabolites expected to be enriched by random chance.

In addition to MPEA, we also sought to ascertain the individual contribution of constituent KEGG metabolites on the ability of DESI-MSI to distinguish between benign and tumour epithelial cells. To do so, we trained a small neural network to distinguish the spectra of epithelial ROIs annotated as either tumour or benign (Fig. 4e). The full dataset consistent of a total of 1163 ROIs ($n = 695$ benign and $n = 468$ tumour) that had also been used for the MPEA. Each sample, corresponding to an ROI, consisted of the mean value of 31 selected metabolites corresponding to KEGG glycolysis and glyconeogenesis, TCA cycle, and fatty acid synthesis metabolic pathways (Supplementary Table 1). As a first step, we normalised the data, such that the mean intensity of each metabolite across all ROIs was equal to 0, with a

standard deviation of 1. A suite of models using 5-fold cross validation were trained. The splits were determined at the patient level, such that no samples from patients in the test set were present in the training set. This was done to ensure that the model learned features which distinguished benign regions from tumour regions and did not memorise patient-specific features common to both tumour and benign regions. The neural network had a multi-layer perceptron architecture, with 64 neurons in each of the two hidden layers. We used ReLu activations in the hidden layers, with a single neuron in the final layer which used a sigmoid activation. We used a binary cross entropy loss function, the Adam optimiser with a learning rate of $10^{-4}$ and we used a batch size of 32. To prevent overfitting, we applied data augmentation during training by adding a small amount of Gaussian noise to each sample. The amplitude of the noise was set to be 10% of the absolute value of each metabolite. In addition, we employed an early stopping protocol to determine the number of epochs for training, based on the validation loss. To evaluate the importance of its constituent features on the final model performance, we combined Shapley additive explanations (SHAP) for each test set in our 5-fold cross validation training scheme[59]. The median performance of the tissue classifier for discriminating between benign and tumour ROI classes across the five different networks was assessed using the area under the ROC curve.

### Open-source spatial transcriptomics data analysis

To evaluate the generalisability of the key findings from the HP-$^{13}$C-MRI cohort, we analysed publicly available mRNA data related to the expression of *SLC16A1, LDHA, LDHB, MPC2, FASN*, and *AR* in benign and tumour surgical specimens. First, bulk mRNA sequencing outputs (log-transformed FPKM values) of the TCGA-PRAD study[43] were accessed through the NCI GDC Data Portal[88] and compared between the two sample classes using the Mann–Whitney *U* test. To allow comparability with the HP-$^{13}$C-MRI cohort, only data from patients with primary Gleason patterns of 3 and 4 were analysed.

In addition, single-cell and spatial RNA-sequencing analyses were performed on two datasets (EGAS00001005787 and GSE176031). For each, data were all processed using the Seurat toolkit (https://satijalab.org/seurat/)[94–97]. Data, obtained as matrices comprising feature (gene) counts as rows and barcoded cells as columns (with annotations regarding cell type and other metadata as available), were first pre-processed by filtering out cells with over 2500 or under 200 unique feature/gene counts as well as those with >5% of counts including mitochondrial DNA. Data were then log-normalised and scaled by a factor of 10,000. Highly variable features were selected using Seurat's 'vst' selection method, and a linear scaling transformation was performed before dimensionality reduction was performed via principal component analysis (PCA). Jackstraw plots were generated[98] to select an appropriate number of dimensions, which was also validated by ranking principal components by the percentage of variance explained by adding each principal component. The "appropriate" number of dimensions was that at which no appreciable increase in signal/percentage of variance explained was obtained by adding an additional dimension. Cells were subsequently clustered using the Louvain clustering algorithm[99], set to a clustering resolution of 0.5. Nonlinear dimensionality reduction was performed with the UMAP method[100]. Differential expression among clusters was analysed using Seurat's default differential expression analysis function, 'FindAllMarkers'. Finally, for analysis of gene signatures comprising multiple genes, a new "meta-feature" was created using Seurat's 'MetaFeature' function, which calculates the relative contribution of a gene set to each cell in the dataset.

For integration of spatial transcriptomics data, 10x Visium data were used and loaded to R as an object comprising an image of the tissue slice along with spot-by-spot gene expression data. As above, data were preprocessed by normalisation to account for spot-by-spot differences in sequencing depth. SCTransform[101] was used to normalise data and detect high-variance features.

### Statistics and reproducibility

Statistical analyses were conducted using GraphPad Prism (version 9.0.2, GraphPad Software, San Diego, CA, USA). Normal distribution of the data was assessed using the D'Agostino-Pearson test (threshold $P \geq 0.05$). All intergroup comparisons were performed using the two-sided Mann–Whitney *U* test or Wilcoxon signed-rank test as appropriate. Correlation analysis was conducted using the Spearman's rank correlation test since at least one variable was always non-normally distributed. In the HP-$^{13}$C-MRI cohort, intergroup comparison of HP-$^{13}$C-MRI-derived [1-$^{13}$C]pyruvate and [1-$^{13}$C]lactate labelling was conducted between $n = 13$ HP-$^{13}$C-MRI-visible tumours (total carbon SNR threshold >5.0 according to the Rose criterion[84,102]) and $n = 13$ contralateral areas of benign tissue. Simultaneously, due to interval androgen deprivation therapy (ADT) administered to patient 5 in Table 1, their two lesions were excluded from intergroup comparison of tissue-based parameters, with the remaining sample size including $n = 11$ tumour and $n = 11$ benign areas. In addition, values of MPC1 and MPC2 epithelial density were log-transformed due to their high range. In the DESI-MSI cohort, the comparison of whole-core epithelial lactate abundance, tissue density, epithelial and stromal cell fractions, as well as compartmentalised lactate pool was made between cores including purely benign ($n = 61$) and >50% tumour ($n = 38$) tissues (Supplementary Fig. 5). MPEA and deep-learning-based tissue classifier were built using the ROIs derived from all tissue cores included in the study, including $n = 23$ mixed cores. For intergroup comparisons and correlation analyses, no multiplicity correction was applied, and therefore all significant tests should be interpreted as exploratory rather than confirmatory. Given the lack of previous clinical studies addressing the key research questions presented in this work, no statistical power calculations could be performed in advance of commencing the study. All experiments were independent and standalone.

### Reporting summary

Further information on research design is available in the Nature Portfolio Reporting Summary linked to this article.

## Data availability

The authors declare that the clinical and imaging data supporting the findings of this study are available within the article and its Supplementary Information. The open-source TCGA-PRAD data used in this study are available through the NCI GDC Data Portal, with additional single-cell and spatial RNA-sequencing analyses performed on two publicly available datasets, EGAS00001005787 and GSE176031, respectively. Data used to generate plots in Figs. 2–7 along with the DESI-MSI metabolite data are provided in the Source Data file. The authors defer raw DESI-MSI and clinical MRI data deposition to ensure compliance with legal requirements of the University of Cambridge and Cambridge University Hospitals NHS Foundation Trust and avoid compromising privacy of the study participants. Requests for raw data can be referred to the corresponding author (N.S.); these will be reviewed within ten working days in consultation with the institutional R&D which will determine the terms of a data transfer agreement between the recipient institution, the University of Cambridge, and Cambridge University Hospitals NHS Foundation Trust. Source data are provided with this paper.

## Code availability

The code to build a metabolic tissue classifier[103] is available at (https://github.com/AstraZeneca/metabolic_classifier). The code to conduct metabolic pathway enrichment analysis is available at (https://github.com/AleksZakirov/MPE-analysis-for-prostate-cancer-study).

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

## Acknowledgements

This study was supported by Prostate Cancer UK (PCUK; Grant PA14-012), Cancer Research UK (CRUK; Grant C19212/A27150), AstraZeneca, and the National Institute for Health and Care Research (NIHR) Cambridge Biomedical Research Centre (BRC; Grant NIHR203312). The views expressed are those of the authors and not necessarily those of the NIHR or the Department of Health and Social Care. N.S. was supported by the Gates Cambridge Trust and is now a Research Fellow of Emmanuel College, Cambridge. A.Y.W. is supported by the Urological Malignancies Programme of the Cancer Research UK Cambridge Centre (C9685/A25177) and NIHR Cambridge Biomedical Research Centre (BRC-1215-20014). V.J.G. acknowledges infrastructure support from the NIHR Cambridge Biomedical Research Centre (BRC-1215-20014). Additional support provided from the Cancer Research UK Cambridge Centre, the Cambridge Experimental Cancer Medicine Centre, a Wellcome Trust Strategic Award, Addenbrooke's Charitable Trust, the Canadian Institute For Advanced Research, and Cambridge University Hospitals National Health Service Foundation Trust. The authors thank Dr Julia Jones of the Cancer Research UK Cambridge Institute for her help with the RNAscope component of the study.

## Author contributions

N.S., G.H., S.T.B., R.J.A.G., T.B., and F.A.G. formulated the research idea, designed and planned the study. N.S., G.H., T.B., and F.A.G. wrote the article. N.S., G.H., L.F., D.B., A.Z., J.R., S.L., J.Y.T., M.A.M., I.G.M., V.A., I.H.M., C.B., J.L.M., I.G.M., and A.Y.W. performed imaging and tissue-based studies and analysed the data. N.S., V.J.G., T.B., and F.A.G. coordinated patient recruitment and oversaw their clinical management. N.S. and G.H. contributed equally to this manuscript. R.J.A.G., T.B., and F.A.G. jointly supervised this work.

## Competing interests

G.H., L.F., D.B., J.R., S.L., J.Y.T., S.T.B., and R.J.A.G. are AstraZeneca employees. F.A.G. has research support from GE Healthcare and AstraZeneca, grants from GlaxoSmithKline, and has consulted for AstraZeneca on behalf of the University of Cambridge. The remaining authors declare no competing interests.
