## [Peer Review File · Nature Communications]

Metabolic imaging across scales reveals distinct prostate cancer phenotypesREVIEWER COMMENTS

Reviewer #1 (Remarks to the Author):

The manuscript describes a study that aims at investigating the underlying biological mechanisms for differences between prostate cancer (PCa) and normal tissue in the observed hyperpolarized (HP) ¹³C lactate after injection of HP ¹³C pyruvate. To this end the authors analyzed data from a cohort of PCa patients that underwent HP MRI as well as conventional MRI (DWI, DCE) prior to prostatectomy with spatially resolved histopathology performed on the resected gland. These results were complemented with data from a second cohort of patients that also underwent radical prostatectomy where desorption electrospray ionization mass spectrometry imaging (DESI-MSI) was used to directly measure endogenous epithelial abundance of multiple metabolites. Patients were selected so that lesions histologically matched those from the HP MRI cohort.

The main findings are:

- Increased HP ¹³C lactate in tumor compared to normal prostate is primarily due to higher epithelial cell density and vascularity rather than differences in epithelial lactate concentration
- HP lactate has strong negative correlation with tumor epithelial MPC2 density, which is also higher compared to healthy tissue
- A subset of lesions of cribriform subtype lacked HP lactate labeling with differential HP lactate labeling of intermediate-risk PCa depending on phenotype

Metabolic imaging of HP pyruvate is an emerging technique that has shown potential for application in detecting and characterizing lesions in patients with prostate cancer, in particular for intermediate-risk disease, for as well as assessing treatment response after therapy. The manuscript provides important information on the underlying biology for the differential HP lactate signal that highlights strengths and limitations for the clinical applications of this technique.

My main concern is with the authors conclusion, highlighted in the abstract, that "tumours of the cribriform subtype may lack [1-¹³C]lactate labelling, which is explained by their lower epithelial lactate dehydrogenase expression, higher mitochondrial pyruvate carrier density, and increased lipid abundance compared to lactate-rich non-cribriform lesions."

There are significant differences with respect to LDH, MPC, lipid abundance as well as epithelial lactate as measured with MSI. However, one of the cribriform lesions, corresponding to 33% of this subtype, showed detectable HP lactate. The comparisons in Fig. 5 are between ¹³C-occult vs ¹³C-visible lesions. It would be useful to highlight the data points from the one ¹³C-visible cribriform lesion in the various plots of Fig. 5. The results with respect to cribriform subtype are important finding that indicates limitations of the imaging technique. But given the small sample size (2/3) it would be more accurate to say that some lesions of cribriform subtype lacked lactate labeling rather than "tumours of the cribriform subtype may lack [1-¹³C]lactate labelling."

The HP MRI data were quantified in terms of pyruvate and lactate SNR. These metrics directly depend on the level of polarization, and based on the data provided in Ref. 2 there was quite a large range in this parameter (7% to 33.8%). These parameters are also directly affected by the sensitivity profile of the receive coil. It would be helpful if the authors would also analyze the HP MRI data in terms of total lactate-to-pyruvate ratio or apparent conversion rate constant k_{PL} , which are both independent on the polarization.

Another aspect that the authors should discuss is the potential of imaging the increased mitochondrial import. The introduction lists "mitochondrial flux as measured through the formation of ¹³C-bicarbonate" as one of the metabolic pathways of [1-¹³C]pyruvate. Did the authors detect any bicarbonate in the tumors or healthy prostate? Could HP bicarbonate be a marker for "increased tumour mitochondrial pyruvate flux"? Please discuss the potential of using HP pyruvate to probe this aspect of altered metabolism in PCa.

Other comments:

1. There are inconsistencies between the intensity scale in the metabolic SNR maps (on the order

of tens of thousands) and scale of the bar graph plots (< 100).

2. Fig. 5 caption: "differential HP-13C-MRI-visibility of cribriform and non-cribriform PCa"

This needs to be revised as comparison is not for cribriform vs non-cribriform lesions.

3. "benign cores were mostly comprised of stromal tissue, whereas tumour samples were predominantly epithelial (P < 0.005 for both; Fig. 2d)."

In the figure it says P=.009 for the comparison in benign cores. Also, "mostly comprised" seems exaggerated as the respective average values are about 40 and 60%.

4. "While neither of these markers (Ktrans,CD31) showed a significant relationship with tumour [1-13C]lactate SNR2 (Supplementary Fig. 2) "

Was there any correlation of Ktrans or CD31 with pyruvate SNR?

5. "Given the absence of HP-13C-MRI signal in most cribriform PCa"

Given the small numbers (2 out of 3 lesions), this statement is too general and should be rephrased.

Reviewer #2 (Remarks to the Author):

The authors demonstrated a cross scale metabolic imaging approach to distinguish prostate cancer subtypes and benign conditions.

The elevated hyperpolarised [1-13C]lactate MRI signal in prostate cancer compared to the benign prostate was associated with increased tumor epithelial cell density and vascularity, rather than differences in epithelial lactate concentration between tumor and normal.

Another demonstration was that tumors of the cribriform subtype may lack [1-13C]lactate labeling, which is explained by their lower epithelial lactate dehydrogenase expression, higher mitochondrial pyruvate carrier density, and increased lipid abundance compared to lactate-rich non-cribriform lesions.

Interesting approach overall but the following should be characterized:

1) The DESI annotations should be benchmarked. How do we know that those metabolic types are not the ones claimed to be the metabolic names? Can they do MALDI and LC/MS type comparisons to further validate their metabolic identities.

2) The cross-scale approach involves MRI with 1-13C and DESI imaging of tissues along with H&E tissues. How do these samples match in 3D space? The tumor is heterogeneous and which tissue section is selected will define the biology that may or may not be the same as in 1-13Clactate MRI images.

3) RNAscope expressions and IHC images of metabolic transporters should be carefully validated. Especially, RNA images do not look convincing. mostly they are low signals and hard to distinguish them from autofluorescent background.

4) The figures are really hard to read and follow. Authors should increase the font and size of the images while decreasing the density of each figure. It is challenging to read the details.

5) Can they show the patient-to-patient variation using visual clustering and direct comparisons and state patient-related details. Could the results be dependent on donor variations?

6) Eight donors is a bit low number. Can the authors increase this to show more statistical confidence and power?

7) IT is confusing to try to match MRI to tissue images? Is there a better way to follow how the cross scale approach from mri metabolism to tissue metabolism are related in the same tissues

and same/different donors.

overall, interesting research direction but the pathology and radiology image fusions should be better demonstrated. and also this work should be framed as bridging the gap between radiomics and pathomics directions.

Reviewer #3 (Remarks to the Author):

quantitative assessment of mitochondrial pyruvate flux using stable isotope tracing in cells or organoids would add objective data. Similarly, functional validation of MPC1 and MPC2 gene knockdown or overexpression experiments will provide definitive evidence on the role of these carriers in metabolic reprogramming and in how pyruvate/lactate in vivo analysis is interpreted. Lipidomic analysis should provide a more comprehensive understanding of the metabolic changes associated with mitochondrial pyruvate flux in terms of TCA reactivation vs the lipogenic phenotype. Extension of the findings to HP MRI using animal models or PDXs, if feasible, would provide the ideal setting to confirm all the findings outlined in this manuscript.

Metabolic imaging across scales identifies distinct prostate cancer phenotypes

(NCOMMS-23-46894) by Sushentsev, N. et al. (2023)

REVIEWER COMMENTS

Reviewer #1 (Remarks to the Author):

The manuscript describes a study that aims at investigating the underlying biological mechanisms for differences between prostate cancer (PCa) and normal tissue in the observed hyperpolarized (HP) ^{13}C lactate after injection of HP ^{13}C pyruvate. To this end the authors analyzed data from a cohort of PCa patients that underwent HP MRI as well as conventional MRI (DWI, DCE) prior to prostatectomy with spatially resolved histopathology performed on the resected gland. These results were complemented with data from a second cohort of patients that also underwent radical prostatectomy where desorption electrospray ionization mass spectrometry imaging (DESI-MSI) was used to directly measure endogenous epithelial abundance of multiple metabolites. Patients were selected so that lesions histologically matched those from the HP MRI cohort.

The main findings are:

- Increased HP ^{13}C lactate in tumor compared to normal prostate is primarily due to higher epithelial cell density and vascularity rather than differences in epithelial lactate concentration
- HP lactate has strong negative correlation with tumor epithelial MPC2 density, which is also higher compared to healthy tissue
- A subset of lesions of cribriform subtype lacked HP lactate labeling with differential HP lactate labeling of intermediate-risk PCa depending on phenotype

Metabolic imaging of HP pyruvate is an emerging technique that has shown potential for application in detecting and characterizing lesions in patients with prostate cancer, in particular for intermediate-risk disease, for as well as assessing treatment response after therapy. The manuscript provides important information on the underlying biology for the differential HP lactate signal that highlights strengths and limitations for the clinical applications of this technique.

We thank the Reviewer for this comprehensive summary and positive feedback.

1. My main concern is with the authors conclusion, highlighted in the abstract, that "tumours of the cribriform subtype may lack $[1-^{13}\text{C}]$ lactate labelling, which is explained by their lower epithelial lactate dehydrogenase expression, higher mitochondrial pyruvate carrier density, and increased lipid abundance compared to lactate-rich non-cribriform lesions."

There are significant differences with respect to LDH, MPC, lipid abundance as well as epithelial lactate as measured with MSI. However, one of the cribriform lesions, corresponding to 33% of this subtype, showed detectable HP lactate. The comparisons in Fig. 5 are between ^{13}C -occult vs ^{13}C -visible lesions. It would be useful to highlight the

data points from the one ^{13}C -visible cribriform lesion in the various plots of Fig. 5. The results with respect to cribriform subtype are important finding that indicates limitations of the imaging technique. But given the small sample size (2/3) it would be more accurate to say that some lesions of cribriform subtype lacked lactate labeling rather than “tumours of the cribriform subtype may lack $[1-^{13}\text{C}]$ lactate labelling.”

We agree with the Reviewer regarding the current wording and have amended the abstract accordingly:

*We also demonstrate that **some** tumours of the cribriform subtype may lack $[1-^{13}\text{C}]$ lactate labelling, which is explained by their lower epithelial lactate dehydrogenase expression, higher mitochondrial pyruvate carrier density, and increased lipid abundance compared to lactate-rich non-cribriform lesions.*

We have also included the data points for the one ^{13}C -visible cribriform lesion in Fig. 5e, with the amended panel presented below. While its epithelial cell density and microvessel density were similar compared to the ^{13}C -occult cribriform tumours, epithelial LDH density was much higher and MPC2 density much lower in the ^{13}C -visible cribriform tumour.

2. The HP MRI data were quantified in terms of pyruvate and lactate SNR. These metrics directly depend on the level of polarization, and based on the data provided in Ref. 2 there was quite a large range in this parameter (7% to 33.8%). These parameters are also directly affected by the sensitivity profile of the receive coil. It would be helpful if the authors would also analyze the HP MRI data in terms of total lactate-to-pyruvate ratio or apparent conversion rate constant k_{PL} , which are both independent on the polarization.

We thank the Reviewer for highlighting this. The choice of lactate SNR as the key metric in this study was dictated by the results of our previous work mentioned by the Reviewer (Ref. 2), where lactate SNR showed significant correlations with histopathological (percentage of Gleason pattern 4 disease) and conventional imaging (tumour cellularity measured as ^1H -MRI-derived ADC) biomarkers of tumour aggressiveness. Considering the clinical relevance of this finding in this specific cohort, we have retained the lactate SNR measurements in the main text. However, given the Reviewer’s comment regarding polarisation levels, we have

now analysed the HP-¹³C-MRI data in terms of k_{PL} . The polarisation range listed is influenced by a small number of outliers and all but three of the imaging studies were undertaken with a polarisation in the range 19-28%. Nonetheless, the k_{PL} results are independent of this as the Reviewer mentions, and the k_{PL} data is now presented in a new Supplementary Fig. 1 (see below). We have also amended the Methods section of the main manuscript to reflect the importance of interpreting the key results in the context of k_{PL} as a metric independent of polarisation:

In addition, we calculated the apparent reaction rate constant for the exchange of the HP ¹³C label between pyruvate and lactate (k_{PL}) using a two-site exchange model using a frequency-domain approach and linear least-squares fitting². k_{PL} quantification was important given the independence of this parameter from the polarisation level, which in this cohort varied between 7%-33.8%, as indicated previously².

In addition, we have amended the Results section of the main manuscript by highlighting the consistency of our lactate SNR findings with those obtained using k_{PL} as follows:

*In the HP-¹³C-MRI cohort, both hyperpolarised [1-¹³C]pyruvate and [1-¹³C]lactate were measured in histopathologically-proven tumour areas ($n = 13$; $P < 0.001$ for both; **Fig. 2b**), which corresponded to local metabolic hotspots on the k_{PL} maps ($P < 0.001$; **Supplementary Fig. 1a**) and were in line with similar findings described in prior clinical reports^{3,4,30}*

*As shown in **Table 1**, **Fig. 5a-c**, and **Supplementary Fig. 1b**, 2/3 ISUP grade group (GG) 3 ICC lesions from the HP-¹³C-MRI cohort had no detectable [1-¹³C]lactate signal or measurable k_{PL} despite sufficient [1-¹³C]pyruvate delivery and restricted diffusion on ¹H-MRI ADC, indicating high cellularity.*

Finally, we have added the following comment to the Discussion section, which indicates directions for future work:

In addition, increasing spatial resolution would also allow us to eliminate unnecessary variability in the [1-¹³C]pyruvate signal derived from the vascular and extracellular compartments, thereby improving the quantification of k_{PL} as a more reliable HP-¹³C-MRI metric that is independent of polarisation.

Supplementary Fig. 1. Assessment of the apparent reaction rate constant for the exchange of the hyperpolarised ^{13}C label between pyruvate and lactate (k_{PL}) in the benign and malignant prostate, as well as in patients with cribriform prostate cancer. a, Representative whole-mount H&E and k_{PL} map from the patient presented in Fig. 2 of the main text showing increased k_{PL} in the tumour region (red outline on the H&E map) compared to the contralateral benign prostate (green outline on the H&E map), with the scatterplot with bars comparing k_{PL} values between the two tissue types. b, Whole-mount H&E and k_{PL} maps from patients presented in Fig. 5 of the main text who harboured large cribriform tumours, two out of three of which showed an undetectable k_{PL} . A mixed box-and-whisker and scatterplot compares k_{PL} values between ^{13}C -occult and ^{13}C -visible lesions, with the k_{PL} value corresponding to the single case of a ^{13}C -visible cribriform tumour depicted in black. c, Scatterplots with bars comparing k_{PL} values between tumours divided based on the percentage (left) and histological subtype (right) of Gleason pattern 4 disease, similar to the analysis presented in Fig. 6 of the main text.

3. Another aspect that the authors should discuss is the potential of imaging the increased mitochondrial import. The introduction lists "mitochondrial flux as measured through the formation of ^{13}C -bicarbonate" as one of the metabolic pathways of $[1-^{13}\text{C}]$ pyruvate. Did the authors detect any bicarbonate in the tumors or healthy prostate? Could HP bicarbonate be a marker for "increased tumour mitochondrial pyruvate flux"? Please discuss the potential of using HP pyruvate to probe this aspect of altered metabolism in PCa.

We agree with the Reviewer that imaging bicarbonate formation could expand the ability of HP- ^{13}C -MRI to visualise altered pyruvate metabolism in prostate cancer, which is highlighted by the apparent relationship between the mitochondrial pyruvate carrier expression and tumour $[1-^{13}\text{C}]$ lactate labelling. In this study, however, we were unable to detect bicarbonate either in the benign or malignant prostatic tissue. While in most cases this was due to the low bicarbonate SNR that prevented meaningful analysis, in some cases it was difficult to robustly assess whether bicarbonate signal was localised to the prostate or arising from the adjacent bowel wall. As these results have revealed the importance of detecting bicarbonate in prostate cancer, we are modifying the sequence used to maximise its detection: this will be undertaken by truncating the time course slightly to retain more polarisation for spectroscopic acquisition at the end of image acquisition, along with utilising high flip angle imaging to increase the bicarbonate SNR. We have now reflected this in the Discussion section of the main text along the following lines:

Importantly, our ability to resolve the mechanism of post-ADT $[1-^{13}\text{C}]$ lactate labelling changes could be further improved by optimising the capacity for imaging hyperpolarised ^{13}C -bicarbonate in the tumour. While in this study we did not reliably detect ^{13}C -bicarbonate formation in the benign and malignant prostate, future work could maximise the detection of bicarbonate by truncating the imaging time to retain sufficient polarisation for magnetic resonance spectroscopy (MRS) following imaging, along with utilising a higher flip angle acquisition to increase the ^{13}C -bicarbonate SNR. If hyperpolarised ^{13}C -bicarbonate is reliably detected in the prostate, this will expand the clinical potential of HP- ^{13}C -MRI considering the important role of altered mitochondrial pyruvate flux in human PCa development.

Other comments:

4. There are inconsistencies between the intensity scale in the metabolic SNR maps (on the order of tens of thousands) and scale of the bar graph plots (< 100).

We thank the Reviewer for highlighting this. We have now amended Figures 2 and 5 accordingly.

5. Fig. 5 caption: "differential HP- ^{13}C -MRI-visibility of cribriform and non-cribriform PCa" This needs to be revised as comparison is not for cribriform vs non-cribriform lesions.

We have now amended the figure caption as follows to reflect the fact that the absence of discernible HP-¹³C-MRI signal in biopsy-proven cribriform tumours helped us identify epithelial LDH and MPC2 densities as the key biological factors underpinning tumour HP-¹³C-MRI visibility:

Comparative assessment of biological factors underpinning HP-¹³C-MRI-visibility of biopsy-proven PCa.

6. "benign cores were mostly comprised of stromal tissue, whereas tumour samples were predominantly epithelial ($P < 0.005$ for both; Fig. 2d)."

In the figure it says $P = .009$ for the comparison in benign cores. Also, "mostly comprised" seems exaggerated as the respective average values are about 40 and 60%.

We have now amended this sentence as follows:

As expected in this age group³⁷, benign cores harboured a significantly higher proportion of stromal tissue, whereas tumour samples were predominantly epithelial ($P < 0.01$ for both; Fig. 2d).

7. "While neither of these markers (K_{trans}, CD31) showed a significant relationship with tumour [1-¹³C]lactate SNR² (Supplementary Fig. 2) "

Was there any correlation of K_{trans} or CD31 with pyruvate SNR?

We have previously reported the lack of a significant correlation between tumour K^{trans} and [1-¹³C]pyruvate SNR in this cohort (Ref. 2; $\rho_s = -0.14$; $P = 0.69$). Here, we also note the absence of a significant relationship between tumour CD31 density and [1-¹³C]pyruvate SNR ($\rho_s = -0.22$; $P = 0.58$). This result can be explained by the fact that low- and intermediate-risk tumours in this cohort had a relatively homogenous vascularity. We have now amended Supplementary Fig. 4 to include a new correlation plot for tumour [1-¹³C]pyruvate SNR and CD31 density, and made the following note in the revised main manuscript text:

Neither of these markers showed a significant relationship with tumour [1-¹³C]pyruvate and [1-¹³C]lactate SNR² (Supplementary Fig. 4) possibly due to the relatively homogeneous vascularity of low- and intermediate-risk tumours in this cohort². However, increased tumour vascularity may further contribute to the [1-¹³C]lactate labelling identified in the malignant compared to the benign prostate.

Supplementary Fig. 4. Relationship between tumour vascular density, perfusion and permeability, and clinical [1-¹³C]lactate or [1-¹³C]pyruvate labelling. **a**, Spearman's correlation plot comparing hyperpolarised tumour [1-¹³C]lactate SNR and K^{trans} on ¹H-MRI. **b**, Spearman's correlation plot comparing hyperpolarised tumour [1-¹³C]lactate SNR and tissue-based CD31 density. Both panels include individual rank correlation coefficients and corresponding P values. **c**, Spearman's correlation plot comparing hyperpolarised tumour [1-¹³C]pyruvate SNR and tissue-based CD31 density. The lack of a significant relationship between tumour [1-¹³C]pyruvate SNR and tumour K^{trans} has been previously reported in this cohort¹ ($\rho_s = -0.14$; $P = 0.69$). All analyses included data from $n = 11$ HP-¹³C-MRI visible lesions.

8. "Given the absence of HP-¹³C-MRI signal in most cribriform PCa"
Given the small numbers (2 out of 3 lesions), this statement is too general and should be rephrased.

We have now rephrased this statement as follows:

Given the absence of HP-¹³C-MRI signal in some cribriform lesions in this cohort, we conducted a granular comparative metabolic analysis of different Gleason pattern glands included in this study.

Reviewer #2 (Remarks to the Author):

The authors demonstrated a cross scale metabolic imaging approach to distinguish prostate cancer subtypes and benign conditions.

The elevated hyperpolarised [1-¹³C]lactate MRI signal in prostate cancer compared to the benign prostate was associated with increased tumor epithelial cell density and vascularity, rather than differences in epithelial lactate concentration between tumor and normal.

Another demonstration was that tumors of the cribriform subtype may lack [1-¹³C]lactate labeling, which is explained by their lower epithelial lactate dehydrogenase expression, higher mitochondrial pyruvate carrier density, and increased lipid abundance compared to lactate-rich non-cribriform lesions.

Interesting approach overall but the following should be characterized:

1. The DESI annotations should be benchmarked. How do we know that those metabolic types are not the ones claimed to be the metabolic names? Can they do MALDI and LC/MS type comparisons to further validate their metabolic identities.

We thank the Reviewer for their feedback and helpful comments. The selected reference m/z values (mapped within 5 ppm accuracy) listed in Supplementary Table 1 are well established and are routinely used in our laboratory. To validate the DESI-MSI results, we have now performed MALDI-MSI (negative ion mode) on serial sections of samples representing benign and malignant prostatic cores, which showed comparable mass accuracy for detecting selected metabolites used in this study (see new Supplementary Table 2 below), along with a similar spatial localisation (see new Supplementary Fig. 11 below, including arachidonic acid and lactate as two example metabolites of interest). In addition, we have clarified the selection of DESI-MSI as the method of choice due to its ability to acquire the data rapidly and with high sensitivity. The amended text of Methods now reads as follows:

*Importantly, DESI-MSI was chosen over other spatial metabolomics techniques, such as matrix-assisted laser desorption ionisation (MALDI) MSI, as it rapidly acquires data and is particularly sensitive for the detection of small molecules which is the focus of this study⁹¹. However, to validate our DESI-MSI findings against an orthogonal technique, we performed MALDI-MSI in negative ion mode on serial sections of samples representing benign and malignant prostatic cores, which showed comparable mass accuracy for metabolite detection (**Supplementary Table 2**) and the key metabolites of interest showed a similar spatial localisation (**Supplementary Fig.11**).*

Metabolite	Pathway	Ion Cluster	m/z calc.	m/z meas. DESI	Error ppm	m/z meas. MALDI	Error ppm
Lactate	TCA cycle	[M-H] ⁻	89.024419	89.0243	0.3	89.0245	0.9
Arachidonic Acid	Fatty acid biosynthesis	[M-H] ⁻	303.232954	303.2337	2.7	303.2322	2.5
Malate	TCA cycle	[M-H] ⁻	133.01425	133.0140	1.8	133.0142	0.4
Citrate	TCA cycle	[M-H] ⁻	191.01973	191.0205	4.0	191.0194	1.7
Palmitoleic acid FA 16:1	Fatty acid biosynthesis	[M-H] ⁻	253.21621	253.2168	2.3	253.2164	0.8
Oleic acid FA 18:1	Fatty acid biosynthesis	[M-H] ⁻	281.247507	281.2489	4.8	281.2486	3.9
Tetradecanoic acid	Fatty acid biosynthesis	[M-H] ⁻	227.2014	227.2020	2.5	227.2009	2.2
Octadecanoic acid	Fatty acid biosynthesis	[M-H] ⁻	283.26539	283.2633	7.3	283.2639	5.3

Supplementary Table 2. *The list of comparative DESI-MSI and MALDI-MSI mass accuracy measurements for selected metabolites-of-interest.*

Supplementary Fig. 11. Direct comparison of DESI-MSI and MALDI-MSI outputs in benign and malignant prostatic cores from the secondary cohort. Representative H&E images of benign and tumour cores are provided, along with corresponding DESI-MSI and MALDI-MSI (negative ion mode) images demonstrating similar spatial distribution and comparable relative abundance of arachidonic acid (m/z 303) and lactate (m/z 89).

2. The cross-scale approach involves MRI with ^{13}C and DESI imaging of tissues along with H&E tissues. How do these samples match in 3D space? The tumor is heterogeneous and which tissue section is selected will define the biology that may or may not be the same as in ^{13}C lactate MRI images.

We thank the Reviewer for highlighting this. HP- ^{13}C -MRI and DESI-MSI were performed in two separate patient cohorts (Fig. 1). In this study, we correlated HP- ^{13}C -MRI, IHC, and RNAscope data derived from large areas of benign and malignant prostate segmented on corresponding whole-mount sections as shown in Supplementary Fig. 5. Matching between HP- ^{13}C -MRI images and whole-mount H&E, IHC, and RNAscope sections was made according to previously described methodology (10.2214/AJR.15.14338). Specifically, whole-mount H&E slices were matched to anatomical T_2 -weighted axial MRI slices in consensus by four of the authors (three urological imagers and one genitourinary pathologist), considering anatomical landmarks, such as prostatic zonal boundaries, urethra, and pseudocapsules of benign hyperplastic nodules if present, while correcting for the distorting effects of specimen preparation where possible. We have now added a new Supplementary Fig. 7 (see next page) to demonstrate the matching of the anatomical structures across HP- ^{13}C -MRI, ^1H -MRI-derived T2WI, and whole-mount H&E sections. Given the accuracy of the image registration, as well as the whole-mount nature of surgical H&E samples and whole-organ MRI images used in the primary cohort, we believe that this approach enabled us to capture heterogeneity of both tumour and benign areas including in this analysis, and profile their metabolism across scales at comparable spatial resolution. To explain this further in the main manuscript, we have amended the Methods section as follows:

*Importantly, to ensure adequate matching of the ROIs drawn on the whole-mount H&E samples, these were matched to the corresponding benign and malignant areas on HP- ^{13}C -MRI: whole-mount H&E slides were co-registered with ^1H -MRI-derived anatomical T_2 -weighted images according to previously described methodology⁹⁰. Specifically, the two types of images were matched by four readers (three urological imagers, N.S., T.B., and F.A.G., and one genitourinary pathologist, A.Y.W.) using anatomical landmarks, such as prostatic zonal boundaries, urethra, and pseudocapsules of benign hyperplastic nodules if present, while correcting for the distorting effects of specimen preparation where possible, as shown in **Supplementary Fig. 7**. This registration, as well as the whole-mount nature of surgical H&E samples and whole-organ MR images, enabled us to capture heterogeneity of both tumour and benign areas including in this analysis.*

DESI-MSI maps were directly fused with the H&E images of the underlying punch biopsies, which enabled us to directly evaluate the metabolic phenotype of benign and malignant glands of interest. To capture tumour heterogeneity, all available tumour vials were included in this study, which has now been clarified in the Methods section as follows:

*To ensure adequate preservation of tumour heterogeneity, all available tumour punch biopsies were included in this study, as shown in **Fig. 1**.*

Supplementary Fig. 7. Anatomical matching of HP-¹³C-MRI, ¹H-MRI, and whole-mount H&E images in the primary cohort. To ensure adequate three-dimensional matching of benign and malignant tissue ROIs included in this analysis involving direct comparison of several imaging modalities, we matched whole-mount H&E slices (right column) with anatomical T₂-weighted images (middle column) as described in the Methods sections of the main text. This figure illustrates matching of the location and shape of such anatomical landmarks as the urethra (white and black arrows) and pseudocapsules of benign hyperplastic nodules (white and black asterisks), as shown in four example cases. Anatomical matching of ¹³C and ¹H-MRI, as illustrated in the image overlay in the left column, was performed simply by aligning the DICOM coordinates. The resulting matched benign and tumour ROIs are presented in **Supplementary Fig. 5**.

3. RNAscope expressions and IHC images of metabolic transporters should be carefully validated. Especially, RNA images do not look convincing. mostly they are low signals and hard to distinguish them from autofluorescent background.

In this study, we used commercially available *LDHA* and *LDHB* RNAscope probes, which are rigorously validated by the manufacturer according to a previously described protocol (10.1016/j.jmoldx.2011.08.002). Here, an experienced member of our Histopathology Core Facility ran the negative control slides (4 Plex DapB to ensure that DapB is in every channel) to assess background staining, along with the positive control slides to determine good RNA quality. In the analysis optimisation, we used the negative controls to set the thresholds for positive signal in the test slides. In addition, the potential impact of autofluorescence was mitigated by another experienced member of our core facility (C.B.) according to the protocol the group has published previously (10.1007/978-1-0716-0623-0_2). We believe that figures in the manuscript demonstrate convincing RNAscope signal, apart from those samples with cribriform disease where low LDH expression was subsequently linked to low endogenous lactate abundance. We have now reflected the above in the Methods section of the main manuscript as follows:

The slides were imaged on the AxioScan (Carl-Zeiss-Stiftung, Stuttgart, Germany) to create whole-slide images. Images were captured at 40x magnification, with a resolution of 0.25 microns per pixel. HALO v3.2.1851.266 and the FISH v2.2.0 modules were used for the automated analysis of scanned RNAscope sections by an experienced member of our Core Facility (C.B.), which included specific steps aimed at overcoming autofluorescence as previously described⁵¹. Importantly, prior to the analysis, we ran the negative control slides (4 Plex DapB to ensure that DapB is in every channel) to assess background staining, along with the positive control slides to determine good RNA quality. In the analysis optimisation, we used the negative controls to set the thresholds for positive signal in the test slides.

The MCT1 antibody has been routinely used in our Histopathology Core Facility since it was validated in a brain tumour cell line and a no primary antibody control in 2014. In this study, we validated antibodies for MPC1 and MPC2 within the same dedicated Histopathology Core Facility (led by J.L.M.) under the supervision of an expert genitourinary pathologist (A.Y.W.). Several antibody titrations were tested in human samples containing healthy renal tissue and renal cell carcinoma. As shown in the corresponding Human Protein Atlas entries, both MPC1 and MPC2 are overexpressed in healthy tubular epithelium compared to RCC epithelium, which was observed during the validation process as shown in a new Supplementary Fig. 9 (below). Importantly, in the prostate samples shown in this study, we observed a reverse trend in line with the Human Protein Atlas findings (prostate cancer epithelium overexpresses both MPC1 and MPC2 compared to the healthy prostatic epithelium), supporting the robustness of these antibodies. We have now amended the Methods section of the main manuscript as follows:

All IHC antibodies used in this study were previously validated in our Histopathology Core Facility as described in the **Reporting Summary**. Here, in addition to routinely used MCT1, HIF-1 α , FASN, and AR antibodies, we additionally validated antibodies for MPC1 and MPC2 within a dedicated Histopathology Core Facility (led by J.L.M.) under the supervision of an expert genitourinary pathologist (A.Y.W.), with further details and representative images from control tissues provided in **Supplementary Fig. 7**.

Supplementary Fig. 9. MPC1 and MPC2 antibody validation. For validation purposes, human kidney samples containing healthy tubular epithelium (left column) and foci of renal cell carcinoma (RCC; right column) were stained with MPC1 and MPC2 antibodies using several antibody titrations. As demonstrated by the relevant Human Protein Atlas entries, both MPC1 and MPC2 are overexpressed in healthy tubular epithelium compared to RCC epithelium, which was observed during the validation process as shown in these representative images. For MPC1, the 1:300 antibody titration was optimal, while for MPC2, the 1:200 antibody titration gave the best contrast between the two tissue types. Scale bars for images of healthy tubular epithelium denote 100 μ m, and 50 μ m for images of RCC epithelium.

4) The figures are really hard to read and follow. Authors should increase the font and size of the images while decreasing the density of each figure. It is challenging to read the details.

Where possible, we have now increased the font in figures throughout both the main text and Supplementary Information. Considering the complexity of the study, the restrictions on the number of illustrative elements permitted for the main text, and the digital format of the Journal, we have retained some of the font sizes, but we are happy to comply with any editorial requirements.

5) Can they show the patient-to-patient variation using visual clustering and direct comparisons and state patient-related details. Could the results be dependent on donor variations?

We have now created a new Supplementary Fig. 2 (below), which demonstrates the patient-to-patient variation of tissue metabolite abundance underpinning the key findings of the manuscript. Panel (a) presents the patient-to-patient variation of whole-core epithelial lactate and palmitoleic acid abundance in benign and tumour samples, with horizontal dotted lines denoting the median values for benign (green) and tumour (red) cores, respectively. As discussed in the main text, there is no clear pattern to explain the differences in epithelial lactate abundance between the two tissue types. In contrast, palmitoleic acid demonstrates increased tumour abundance compared to the benign samples (data are shown in the same panel).

Importantly, when patients were grouped by the amount and phenotype of Gleason pattern 4 disease (see panel (b), below), both lactate and palmitoleic acid showed a consistent pattern, which was not influenced by individual patient variations. Specifically, patients with cribriform tumours clearly showed an increased lactate abundance and decreased palmitoleic acid abundance, while non-cribriform lesions showed the reverse trend. We have now added references to this new figure in the main manuscript as appropriate, with individual patient and tumour characteristics provided in Table 1 of the main manuscript.

Supplementary Fig. 2. Visual representation of patient-to-patient variation in DESI-MSI-derived epithelial lactate and palmitoleic acid abundances in the benign and malignant prostate, as well as lesions of different histological phenotypes. *a*, Mixed box-and-whisker and scatterplots demonstrating patient-to-patient variation of whole-core epithelial lactate and palmitoleic acid abundances (both corrected for cell density), with green and red horizontal lines denoting median values corresponding to all benign and malignant samples, respectively. *b*, Mixed box-and-whisker and scatterplots presenting patient-to-patient variation of the same metabolite abundances, with tumours grouped as those with %GP4 < 10 (orange), %GP4 > 10 non-cribriform (NC; red), and %GP4 > 10 invasive cribriform carcinoma (ICC; black), with corresponding median values presented as horizontal lines. Individual patient and lesion characteristics are summarised in Table 1 of the main text, with the figure used to support the key findings presented in Fig. 2 and Fig. 7 of the main text.

6) Eight donors is a bit low number. Can the authors increase this to show more statistical confidence and power?

We agree with the Reviewer that the limited sample size of the HP-¹³C-MRI cohort is an important limitation of this study, which will be addressed in the future work as stated in the limitations section of the Discussion. However, recruiting additional patients to this prospective study for this specific paper is not feasible for a number of reasons outlined below. The MISSION-Prostate MRI study had to be ended prematurely in March 2020 when all prospective clinical studies were halted by the COVID-19 pandemic. Hence, we are now focused on delivering other prospective studies that involve patients who do not undergo radical prostatectomy, which is central to the adequate matching of imaging data with the underlying tissue biology as highlighted by this Reviewer (Comment 2). Moreover, the current sample size of the HP-¹³C-MRI cohort (8 patients, 15 lesions) is in keeping with that of recent reports published by the other major centres that have the capacity for performing clinical HP-¹³C-MRI in prostate cancer patients (10.1016/j.cmet.2019.08.024; 10.1002/jmri.28467; 10.1038/s41391-019-0180-z). Furthermore, as no prior clinical studies have addressed the research questions presented in this work (i.e. comparison of HP-¹³C-MRI-derived metrics between the benign and malignant prostate and lesions of cribriform and non-cribriform morphology), it was not possible to perform any statistical power calculations in advance of commencing the study, with the work considered to be hypothesis generating. Nevertheless, all key comparisons that we have presented yielded statistically significant results, but have been highlighted as exploratory, as stated in the Statistical Analysis section of this manuscript. We have now amended this section to reflect the above as follows:

Given the lack of previous clinical studies addressing the key research questions presented in this work, no statistical power calculations could be performed in advance of commencing the study.

The Limitations section of the Discussion has also been amended to highlight the need to increase the sample size of future multicentre studies involving HP-¹³C-MRI:

While the HP-¹³C-MRI cohort had a limited sample size, this was in keeping with recent clinical reports by other centres that have the capacity to perform clinical HP-¹³C-MRI studies in patients with PCa^{4,30,84}. In addition, this study did not include patients with GP5 disease or IDC as these rarely undergo surgical treatment, which will be assessed in future work. Importantly, future studies will include larger patient cohorts to assess the generalisability of our exploratory findings and can be statistical powered based on the results here. In addition, future studies can also investigate the role of other metabolic pathways in differentiating PCa subtypes.

7) IT is confusing to try to match MRI to tissue images? Is there a better way to follow how the cross scale approach from mri metabolism to tissue metabolism are related in the same tissues and same/different donors.

As highlighted in our response to Comment 2, the thorough matching of HP-¹³C-MRI, ¹H-MRI, and whole-mount H&E slides in the primary cohort has enabled us to confidently profile metabolism of the same tissue areas across multiple scales. However, alternative methods aimed at ensuring adequate matching between targeted biopsy specimens and HP-¹³C-MRI annotations are required in future work involving non-surgical patient populations. One potential solution may be to place fiducial markers at the time of the targeted biopsy, which would be visible at the subsequent HP-¹³C-MRI scans. We have now highlighted this important avenue for future work in the Discussion section of this manuscript as follows:

Finally, while this study relied on whole-mount sections for three-dimensional matching between clinical HP-¹³C-MRI and tissue-based digital pathology, IHC, and spatial transcriptomics, future work in non-surgical patient populations should consider alternative approaches (e.g. fiducial marker placement) to ensure that targeted biopsy samples truly represent the tissue biology behind HP-¹³C-MRI annotations.

overall, interesting research direction but the pathology and radiology image fusions should be better demonstrated. and also this work should be framed as bridging the gap between radiomics and pathomics directions.

We thank the Reviewer for their feedback and have now addressed their concerns regarding pathology and radiology image fusion in response to Comment 2.

Reviewer #3 (Remarks to the Author):

1. quantitative assessment of mitochondrial pyruvate flux using stable isotope tracing in cells or organoids would add objective data. Similarly, functional validation of MPC1 and MPC2 gene knockdown or overexpression experiments will provide definitive evidence on the role of these carriers in metabolic reprogramming and in how pyruvate/lactate in vivo analysis is interpreted. Lipidomic analysis should provide a more comprehensive understanding of the metabolic changes associated with mitochondrial pyruvate flux in terms of TCA reactivation vs the lipogenic phenotype. Extension of the findings to HP MRI using animal models or PDXs, if feasible, would provide the ideal setting to confirm all the findings outlined in this manuscript.

We thank the Reviewer for their useful suggestions, which would indeed help to confirm and mechanistically explain the key findings of our work. However, due to the current clinical focus of our group, setting up the required facilities for the described experiments would take a significant amount of time and potentially represents a separate programme of work. Hence, while these excellent suggestions are outside the scope of work for the current manuscript at this stage, we intend to pursue future work along the described lines, which we have now incorporated in the Discussion section of our manuscript:

Moreover, while functional validation of the impact of MPC expression on mitochondrial pyruvate flux in PCa cells was outside of the scope of this clinical study, future work using stable isotope tracing in cell lines, organoids, or patient-derived xenografts (PDXs) of cribriform and non-cribriform disease would be key for confirming and explaining the mechanisms underlying the main findings outlined in this study⁸⁵. While similar work has been conducted in cardiomyocytes⁵⁵, taking this forward in the PCa setting along with advanced lipidomic analysis could provide a more comprehensive understanding of the metabolic changes associated with mitochondrial pyruvate flux in terms of TCA reactivation compared to the lipogenic phenotype.

REVIEWER COMMENTS

Reviewer #1 (Remarks to the Author):

The following items should be addressed before publication:

1. Addressing the issue of detecting the conversion of ^{13}C pyruvate to bicarbonate (as a measure of mitochondrial import" the authors responded "future work could maximise the detection of bicarbonate by truncating the imaging time to retain sufficient polarisation for magnetic resonance spectroscopy (MRS) following imaging, along with utilising a higher flip angle acquisition to increase the ^{13}C -bicarbonate SNR.

I'm not quite sure I understand this. Would the goal be to detect bicarbonate with a separate MRS acquisition, rather than optimizing the metabolic imaging sequence for improved detection of bicarbonate? If it is done by MRS, is it possible to differentiate signal from lesion vs normal prostate? Please clarify.

2. Suppl. Fig. 1. There are inconsistencies between the intensity scale in the kPL maps (maximum of 0.25 1/s) and scale of the bar graph plots (maximum of 0.02 1/s).

3. Two of the 3 whole-mount H&E maps Suppl. Fig. 1b appear to be the same as in Fig. 5b and 5c but the third is slightly different from Fig. 5a. Is the corresponding kPL map from a different slice than the Lac SNR map in Fig. 5a?

Reviewer #2 (Remarks to the Author):

Authors addressed most of the concerns regarding pathomics and radiomics fusion. The remaining issues are:

1) While supplementary table 2 shows the similarity of peaks between DESI and MALDI, the supplementary fig. 11 presented different spatial patterns of DESI vs MALDI. The authors should use spatial quantitative methods to minimize the difference between the two images - why do they look like huge variations? can this be minimized using better sample preparation?

2) The validation and benchmarking of IHC and RNAscope markers should not rely on company specs or other publications. The imaging community gets very different results that are not similar to the company's claims. Thus, authors should use positive and negative controls to validate their IHC and RNAscope stains. This is extremely important to benchmark their findings for metabolic correlations across multi-scales.

3) GitHub links should be provided to the reviewers for a complete understanding of the accuracy of the computational methods implemented here.

Reviewer #2 (Remarks on code availability):

The manuscript mentions "The code to build a metabolic tissue classifier is available at (link to GitHub to be provided upon publication). The code to conduct metabolic pathway enrichment analysis is available at (link to GitHub to be provided upon publication)"

However, this statement is not acceptable for reviewers - how do we know their codes are properly implemented?

Metabolic imaging across scales identifies distinct prostate cancer phenotypes
(NCOMMS-23-46894A) by Sushentsev, N. et al. (2023)

REVIEWER COMMENTS

REVIEWER COMMENTS

Reviewer #1 (Remarks to the Author):

The following items should be addressed before publication:

1. Addressing the issue of detecting the conversion of ^{13}C pyruvate to bicarbonate (as a measure of mitochondrial import" the authors responded "future work could maximise the detection of bicarbonate by truncating the imaging time to retain sufficient polarisation for magnetic resonance spectroscopy (MRS) following imaging, along with utilising a higher flip angle acquisition to increase the ^{13}C -bicarbonate SNR.

I'm not quite sure I understand this. Would the goal be to detect bicarbonate with a separate MRS acquisition, rather than optimizing the metabolic imaging sequence for improved detection of bicarbonate? If it is done by MRS, is it possible to differentiate signal from lesion vs normal prostate? Please clarify.

The ideal approach would be to detect ^{13}C -bicarbonate using an optimised sequence for evaluating both $[1-^{13}\text{C}]$ lactate and ^{13}C -bicarbonate within the prostate. However, detecting ^{13}C -bicarbonate is challenging even in the brain, where we know bicarbonate production is relatively high (doi: 10.1016/j.neuroimage.2019.01.027; doi: 10.1148/rycan.210076). We have obtained more reliable ^{13}C -bicarbonate readings in the brain using MRSI (doi: 10.1016/j.neuroimage.2022.119284), however, this was undertaken at the cost of obtaining a dynamic timecourse, since MRSI uses up a large proportion of the available polarisation. In the prostate, performing slice-localised MRS on regions with and without tumour at the end of a dynamic timecourse would be preferable in this context, allowing both to be obtained from the available polarisation. While this would not allow us to definitively assess ^{13}C -bicarbonate localisation within the slice, our proposal was to use this data to determine the likelihood of detecting ^{13}C -bicarbonate from subsequent dedicated acquisitions that would acquire signal from multiple metabolites simultaneously.

An alternative solution would be to use 3D echo-planar spectroscopic imaging (EPSI), which theoretically allows aliasing of ^{13}C -bicarbonate signal at 3T (doi: 10.1002/nbm.4280), as it is outside the spectral window normally acquired. In practice, however, EPSI-derived ^{13}C -bicarbonate assessment in the prostate where its natural abundance is much lower than in the brain, may be suboptimal due to a decreased signal-to-noise ratio (SNR) compared to MRSI. This may explain the lack of published data demonstrating hyperpolarised ^{13}C -bicarbonate signal in the human prostate. Overall, optimising the metabolic imaging sequence for improved detection of ^{13}C -bicarbonate is an important area for technological development, and there has been some very recent development in this area (10.1021/acssensors.3c008), published since

the original submission of our manuscript. We have now clarified this in the amended version of the manuscript:

While in this study we did not reliably detect ^{13}C -bicarbonate formation in the benign and malignant prostate, future work could maximise the detection of bicarbonate by truncating the imaging time to retain sufficient polarisation to allow for slice-localised magnetic resonance spectroscopy (MRS) following imaging, along with utilising a higher flip angle acquisition to increase the ^{13}C -bicarbonate SNR. We are not aware of any published clinical study reporting the detection of ^{13}C -bicarbonate in the human prostate, and therefore detecting it using MRS would provide evidence to optimise existing spiral or echo-planar spectroscopic imaging (EPSI) sequences in future studies. Important work in this area could also be facilitated by the recently published clinically translatable method of ^{13}C -bicarbonate imaging that showed good performance in preclinical models of PCa⁶⁷. Successfully translating these efforts into clinical studies will expand the clinical potential of HP- ^{13}C -MRI for assessment of altered mitochondrial pyruvate flux in human PCa development. In addition, increasing spatial resolution would also allow us to eliminate unnecessary variability in the $[1-^{13}\text{C}]$ pyruvate signal derived from the vascular and extracellular compartments, thereby improving the quantification of k_{PL} as a more reliable HP- ^{13}C -MRI metric that is independent of polarization levels.

2. Suppl. Fig. 1. There are inconsistencies between the intensity scale in the kPL maps (maximum of 0.25 1/s) and scale of the bar graph plots (maximum of 0.02 1/s).

This has now been amended, with the revised version of Supplementary Fig. 1 provided below.

3. Two of the 3 whole-mount H&E maps Suppl. Fig. 1b appear to be the same as in Fig. 5b and 5c but the third is slightly different from Fig. 5a. Is the corresponding kPL map from a different slice than the Lac SNR map in Fig. 5a?

We thank the reviewer for highlighting this. While most whole-mount sections could be placed on one mega slide (such as those presented in Fig. 5b,c and Supplementary Fig. 1a), some prostate mega blocks had to be split into two, with the resulting whole-mount sections being a fusion of two halves of the prostate. This was the case with the whole-mount section highlighted by the reviewer; to better demonstrate the spatial matching between HP- ^{13}C -MRI and H&E maps (see Supplementary Fig. 7) we re-aligned two halves of the whole-mount section to focus on matched BPH nodules instead of the urethra, which is not as clearly seen on this specific ^1H -MRI slice. We can confirm that both $[1-^{13}\text{C}]$ lactate and k_{PL} maps have been taken from the same slice, which corresponds to this whole-mount section. We have now amended Supplementary Fig. 1b to ensure the consistency of the whole-mount H&E maps with those presented in Fig. 5a of the main text, as well as Supplementary Fig. 7. We have also made a note in the figure legend to highlight that the whole-mount H&E map in question represents a fusion of two separate slides.

Supplementary Fig. 1. Assessment of the apparent reaction rate constant for the exchange of the hyperpolarised ^{13}C label between pyruvate and lactate (k_{PL}) in the benign and malignant prostate, as well as in patients with cribriform prostate cancer. **a**, Representative whole-mount H&E and k_{PL} map from the patient presented in **Fig. 2** of the main text showing increased k_{PL} in the tumour region (red outline on the H&E map) compared to the contralateral benign prostate (green outline on the H&E map), with the scatterplot with bars comparing k_{PL} values between the two tissue types. **b**, Whole-mount H&E and k_{PL} maps from patients presented in **Fig. 5** of the main text who harboured large cribriform tumours, two out of three of which showed an unmeasurable k_{PL} . A mixed box-and-whisker and scatterplot compares k_{PL} values between ^{13}C -occult and ^{13}C -visible lesions, with the k_{PL} value corresponding to the single case of a ^{13}C -visible cribriform tumour depicted in black. **c**, Scatterplots with bars comparing k_{PL} values between tumours divided based on the percentage (left) and histological subtype (right) of Gleason pattern 4 disease, similar to the analysis presented in **Fig. 6** of the main text. In **b**, the whole-mount H&E map on the left is a fusion of two separate slides, which individually represent two halves of the whole gland sectioned at the same level.

Reviewer #2 (Remarks to the Author):

Authors addressed most of the concerns regarding pathomics and radiomics fusion. The remaining issues are:

1) While supplementary table 2 shows the similarity of peaks between DESI and MALDI, the supplementary fig. 11 presented different spatial patterns of DESI vs MALDI. The authors should use spatial quantitative methods to minimize the difference between the two images - why do they look like huge variations? can this be minimized using better sample preparation?

We thank the Reviewer for highlighting this. The apparent differences in spatial patterns of DESI-MSI and MALDI-MSI are representative of the ion suppression effect, which is a well-described phenomenon in MSI resulting from an intricate interplay of physicochemical properties of complex tissues. As demonstrated by Taylor *et al.* (doi: 10.1021/acs.analchem.7b05005), MALDI-MSI is significantly more prone to the ion suppression effect compared to DESI-MSI, which is one of the reasons why we relied on the latter technique in this study. As previously described by Hamm *et al.* (doi: 10.1016/j.jprot.2012.07.035; Dr Hamm is the joint first author of this submission and Director of Imaging & Data Analytics Team at AstraZeneca with a responsibility for managing the MSI facilities), the ion suppression effect in MALDI-MSI is most pronounced between tissues of varying histological and biophysical properties. In Supplementary Fig. 11, the spatial distribution of both arachidonic acid and lactate on MALDI-MSI is almost exclusively limited to tumour epithelium (outlined in red in the revised figure), which is characterised by increased stiffness, tissue fluidity, and cell density (doi: 10.1097/RLI.0000000000000685) compared to the benign epithelium. Simultaneously, DESI-MSI clearly detects the two metabolites in both benign and malignant prostatic epithelium, which is in keeping with its lower susceptibility to the ion suppression effect.

Moreover, unlike DESI-MSI, MALDI-MSI highly depends on the matrix deposition and its properties, which is another factor that can hinder robust qualitative and quantitative metabolite identification. This explains the higher sensitivity of DESI-MSI compared to MALDI-MSI, which was another reason for selecting the former technique in this study. In addition, MALDI-MSI and DESI-MSI use different normalisation methods (TIC and RMS, respectively), which are dependent on mass analysers coupled to each ionisation source (Q-ToF and Orbitrap, respectively).

Finally, the adjustment of signal intensities on MALDI-MSI and DESI-MSI images cannot be executed perfectly in the Scils software. Therefore, since the direct comparison between the two techniques is not possible, the spatial differences described by the Reviewer are expected and have now been explained in the revised legend to Supplementary Fig. 11. Correcting for the differences between the two techniques is possible by applying a labelled standard homogeneously on tissue sections between the MSI experiment; the intensity of detected analytes will be subsequently normalised to this standard on the resulting images. While this approach has been used in quantitative MSI studies targeting exogenous compounds such as drugs, its application to exploratory biological studies is limited due to the untargeted

nature of metabolomic analysis in this case. In conclusion, the use of DESI-MSI in this study is justified by its high sensitivity, reduced susceptibility to the ion suppression effect, and lack of matrix effects, with MALDI-MSI-derived metabolic assessment being outside the scope of this work.

Supplementary Fig. 11. Direct comparison of DESI-MSI and MALDI-MSI outputs in benign and malignant prostatic cores from the secondary cohort. Representative H&E images of benign and tumour cores are provided, along with corresponding DESI-MSI and MALDI-MSI (negative ion mode) images demonstrating similar spatial distribution and comparable relative abundance of arachidonic acid (m/z 303) and lactate (m/z 89). The slight spatial differences between DESI-MSI and MALDI-MSI, best illustrated by the predominant localisation of both metabolites to tumour epithelial glands (red) compared to adjacent benign epithelium (green) can be explained by the ion suppression effect, which is primarily indicative of different biophysical properties of the two tissue types (rather their differential metabolite abundance) and is more pronounced in MALDI-MSI compared to DESI-MSI^{3,4}. Direct comparison between the two techniques is further complicated by the matrix effects specific to MALDI-MSI, different normalisation methods used for the two mass analysers, and imperfect adjustment of signal intensities in the Scils software.

2) The validation and benchmarking of IHC and RNAscope markers should not rely on company specs or other publications. The imaging community gets very different results that are not similar to the company's claims. Thus, authors should use positive and negative controls to validate their IHC and RNAscope stains. This is extremely important to benchmark their findings for metabolic correlations across multi-scales.

We agree with the Reviewer that rigorous validation is essential for the validity and generalisability of biomedical research. We would like to clarify that both IHC and RNAscope components of this study were run by expert technicians who are members of the high-throughput Histopathology/ISH Core Facility embedded into the Cancer Research UK Cambridge Institute. As mentioned in the Methods section manuscript and detailed in the Reporting Summary, all IHC and RNAscope antibodies used in this study were validated in-house using positive and negative tissue controls.

For IHC, antibody validation was performed by Ms Jodi L. Miller (co-author of this submission and Principal Scientific Associate at the Histopathology/ISH Core Facility with more than 10 years' experience of running IHC for more than 100 projects involving a range of more than 250 validated antibodies) as follows:

- **For MCT1, the antibody was initially tested in a brain cell line that served as a positive control. We tested the antibody at a single dilution (1:100) with three different pre-treatments (sodium citrate, Tris EDTA, and enzyme), along with a no primary antibody control for each retrieval. We saw strong cell surface signal with both sodium citrate and Tris EDTA, as expected given that MCT1 is most commonly localised to cell membrane: predominant cell membrane staining in the prostate samples can be clearly seen in Fig. 2i and Fig. 5d of the main text. The initial tissue tests of the antibody were subsequently carried out in a human xenograft rat model (Rat 3(2)_SP13) with a range of antibody titrations. Following the review of positive and negative control sections by a consultant neuropathologist, the antibody dilution to 23.36 µg/mL was deemed optimal; this was subsequently confirmed in primary human tissue. The antibody has been used routinely since it was validated in 2014.**
- **For CD31, we tested the antibody at a single dilution (1:50) with three different pre-treatments (sodium citrate, Tris EDTA, and enzyme) in FFPE human tonsil, along with a no primary antibody control for each retrieval condition. Sodium citrate pre-treatment was taken forward and tested on FFPE human breast cancer TMA offcuts. Results were reviewed by an expert human breast pathologist, with optimal conditions (of a 1:50 antibody dilution and sodium citrate antigen retrieval buffer) agreed. The antibody has been used routinely since it was validated in 2018. Specific vascular staining is clearly demonstrated in Fig. 2i and Fig. 5d on the main text.**
- **For FASN, we tested the antibody at a single dilution (1:100) with three different pre-treatments (sodium citrate, Tris EDTA, and enzyme) in C42b (a FAS over-expressing) and a FAS knockdown FFPE human cell-line, with a no primary antibody control for each retrieval. Sodium citrate and Tris EDTA antigen retrieval yielded positive staining so both antigen retrieval conditions were tested in FFPE human prostate TMA off-cuts next, again including a no**

primary control for each retrieval condition. Tris EDTA was settled upon as giving the best result with positive staining in prostate cancer tissue and negative staining in normal prostate. The antibody has been used routinely since it was validated in 2011. Cytosolic staining is clearly demonstrated in Fig. 3b, Fig. 5d, and Fig. 6c of the main text.

- For AR, we tested the antibody at two dilutions (1:100 & 1:250) with two different pre-treatments (sodium citrate & Tris EDTA) in FFPE LNCaps (a human prostate cell line), along with a no primary control for each retrieval condition. Tris EDTA pre-treatment only was taken forward with a 1:50, 1:100 & 1:250 dilution on FFPE human prostate cancer TMA offcuts, alongside the LNCaps, and with no primary controls. Optimal conditions of a 1:50 antibody dilution and Tris EDTA antigen retrieval were settled upon in agreement with an expert genitourinary pathologist. The antibody has been used routinely since it was validated in 2008. Exclusively nuclear staining is clearly demonstrated in Fig. 3b, Fig. 5d, and Fig. 6c of the main text. Importantly, in Fig. 5b, nuclear AR staining is exclusively seen in luminal epithelial cells in the benign gland, with no staining detected in basal epithelium.
- For HIF-1 α , we tested the antibody at a single dilution (1:100) with three different pre-treatments (sodium citrate, Tris EDTA, and enzyme) in FFPE human breast tissue (normal versus tumour), with a no primary control for each retrieval. Sodium citrate and Tris EDTA antigen retrieval yielded positive staining so both antigen retrieval conditions were tested in FFPE human breast TMA off-cuts next using a dilution of 1:50 & 1:100 for sodium citrate and 1:100 & 1:200 for Tris EDTA. Results were reviewed by an expert human breast pathologist and the antibody dilution of 1:100 (23.36 μ g/mL) with sodium citrate retrieval deemed optimal as it differentiated positive and negative cases clearly. The antibody has been used routinely since it was validated in 2018. Both cytosolic and nuclear staining is clearly demonstrated in Fig. 6c of the main text.
- For MPC1 and MPC2, the validation process has already been described in the Supplementary Information. Granular cytosolic staining is clearly demonstrated in Fig. 3b, Fig. 5d, and Fig. 6c of the main text.

This information has now been included in the revised version of the Reporting Summary document.

For RNAscope, probe validation was performed by Mrs Julia Jones (Scientific Manager at the Histopathology/ISH Core Facility with 11-years of experience in running RNAscope automated kits available for the Leica Bond Rx, as well as manual HiPlex kits, for more than 50 separate projects in a variety of tissues and species). Prior to the analysis, spare tissue sections were used to run the negative control slides (4 Plex DapB to ensure that DapB is in every channel) to assess background staining, along with the positive control slides (*POLR2A* for channel 1 and *PPIB* for channel 2) to determine good RNA quality. In the analysis optimisation, we used the negative controls to set the thresholds for positive signal in the test slides. This information has now been included in the Methods section of the main manuscript as follows:

In addition, prior to the analysis, we used spare tissue sections to run the negative control slides (4 Plex DapB to ensure that DapB is in every channel) to assess background staining, along with the positive control slides (POLR2A for channel 1 and PPIB for channel 2) to determine good RNA quality. In the analysis optimisation, we used the negative controls to set the thresholds for positive signal in the test slides. The described routine in-house RNAscope antibody validation process, along with the subsequent analysis, was performed by an experienced member of our dedicated Histopathology Core Facility (see **Acknowledgments**) with 11-years of experience of using RNAscope automated kits available for the Leica Bond Rx (Single Plex, Duplex, 3 Plex, 4 Plex, BaseScope, and RNAscope Plus), as well as manual HiPlex kits, for more than 50 separate projects in a variety of tissues and species, including human and murine breast, brain, kidney, lung, and liver.

Given the volume of RNAscope projects run at the Histopathology/ISH Core Facility, scanned control images for this project (completed in 2020) were no longer stored in the system. However, we have more recently validated *LDHA* and *LDHB* RNAscope probes for projects involving human renal (panel A below) and breast (panel B below) tissues following the same routine procedure. For the reviewer's reference, the figure below demonstrates scanned images of FFPE specimens from the two validation experiments. In both experiments, the negative control slides had DapB in every channel, while the positive control slides included *POLR2A* and *PPIB* probes. In panel A, fluorescent scanning was performed using Akoya Phenolmager HT, with subsequent images unmixed on InForm software to separate autofluorescence. In panel B, images were acquired using Zeiss widefield microscope, with the bottom sub-panel demonstrating the outputs of a classifier used to filter out autofluorescence.

A

B

3) GitHub links should be provided to the reviewers for a complete understanding of the accuracy of the computational methods implemented here.

Reviewer #2 (Remarks on code availability):

The manuscript mentions "The code to build a metabolic tissue classifier is available at (link to GitHub to be provided upon publication). The code to conduct metabolic pathway enrichment analysis is available at (link to GitHub to be provided upon publication)" However, this statement is not acceptable for reviewers - how do we know their codes are properly implemented?

At the request of the Editor following the initial submission of our manuscript, we had provided the Software Policy checklist along with with a compressed .zip file containing all the software/code used in this study. The .zip file, including example data and other relevant instructions to check the code implementation, was also attached to the revised version of this submission. We hope that this file, which we have again attached to this revision, can be made accessible to the Reviewers. In the meantime, we are in the process of creating a dedicated GitHub repository for this study, which will include the metabolic classifier code; this will be made publicly available following our internal approval process; the link will be provided in the relevant section of the manuscript should it be accepted for publication. The link to the code for metabolic pathway enrichment analysis is already available at: <https://github.com/AleksZakirov/MPE-analysis-for-prostate-cancer-study>

REVIEWER COMMENTS

Reviewer #2 (Remarks to the Author):

Thanks to the authors for their revision. The remaining concerns include:

1) DESI and MALDI disagreement is understood. I was wondering if there is any pure sample (target or phantom) to convince the readers that the results are consistent.

2) The IHC and RNAscope validations were detailed. Can the authors present positive and negative control data like the LDHA one? These are helpful to have in the supplementary.

the study is interesting and the team is strong but, for reviewers and authors, validation datasets should still be presented beyond presenting experience level of individuals.

Reviewer #2 (Remarks on code availability):

metabolic_classifier.ipynb files should also be provided in GitHub instead of an attachment here.

Metabolic imaging across scales identifies distinct prostate cancer phenotypes
(NCOMMS-23-46894B) by Sushentsev, N. et al. (2023)

REVIEWER COMMENTS

Reviewer #2 (Remarks to the Author):

Thanks to the authors for their revision. The remaining concerns include:

1) DESI and MALDI disagreement is understood. I was wondering if there is any pure sample (target or phantom) to convince the readers that the results are consistent.

To further validate the ability of DESI-MSI to accurately detect targeted metabolites in the prostate tissue samples, we benchmarked its performance against pure metabolite standards using an additional method, tandem mass spectrometry (MS/MS). As shown in the new Supplementary Fig. 13 which is presented below, standards for lactate (a) and arachidonic acid (b) were applied to a target slide and a DESI MS/MS spectrum (negative ion mode) was acquired using collision-induced dissociation (CID) with a nominal collision energy (eV) adjusted for each metabolite and condition. Parent m/z values observed in the standards were subsequently evaluated in the human prostate tissue sections (c, d). Green arrows in each spectrum indicate characteristic fragment peaks that were within a similar range between the control standards and the human prostate test samples, thereby providing further evidence of the reliability of the DESI-MSI derived measurements used in this study. Taken together with the MALDI-MSI derived validation undertaken previously using similar m/z values, we believe that this additional step provides sufficient evidence for the validity of the DESI-MSI results presented in this manuscript. We have now amended the Methods and Discussion sections of the main text as follows:

*In addition, we used tandem mass spectrometry (MS/MS)⁹⁷ to further validate the DESI-MSI derived targeted metabolite detection against metabolite standards. As shown in **Supplementary Fig. 13**, similar characteristic metabolite fragment peaks were observed in both pure control standards and human prostate test samples used in this study.*

Notably, the relatively lower intensity of the parent peak and associated higher background MS/MS spectra are in keeping with the lower endogenous abundance of lactate (c) compared to arachidonic acid (d) in the human prostate. A small metabolite, lactate is also much more difficult to fragment compared to a fatty acid. Hence, these findings provide additional evidence in favour of developing metabolic imaging tools targeting lipid metabolism, rather than probing the abundance of small and less abundant metabolites such as lactate. The revised Discussion section now reflects this as follows:

Conversely, the results here suggest that future development of novel ^{13}C -labelled probes targeting fatty acid metabolism may yield important results given the continuous upregulation of lipid metabolism noted throughout PCa development^{74,75} and the subsequent increase in endogenous fatty acid abundance compared to small metabolites such as lactate.

Supplementary Fig. 13. Confirmation of lactate and arachidonic acid (AA) detection in human prostate tissues using analytical standards. Standards for lactate (a), and AA (b) were applied to a target slide and a DESI MS/MS spectrum (negative ion mode) was collected using collision-induced dissociation (CID) with a nominal collision energy (eV) adjusted for each metabolite and condition. Parent m/z values observed in the standards were subsequently evaluated in the human prostate tissue sections (c, d). Metabolite fragments observed in the tissue were compared to those obtained from the standards. Green arrows in each spectrum indicate characteristic fragment peaks that were similar between the standards and human prostate samples. Identification of fragments from small metabolites such as lactate can be challenging due to the lower intensity of the parent peak and the limited fragmentation pattern on a high background MS/MS spectrum as demonstrated in the prostate specimen in panel (c). This contrasts with the large peak from a large and abundant metabolite such as AA, which demonstrates a higher signal-to-noise ratio on a low background MS/MS spectrum as seen in panel (d).

2) The IHC and RNAscope validations were detailed. Can the authors present positive and negative control data like the LDHA one? These are helpful to have in the supplementary.

the study is interesting and the team is strong but, for reviewers and authors, validation datasets should still be presented beyond presenting experience level of individuals.

As requested, we have now included the LDHA and LDHB positive and negative control data as the new Supplementary Fig. 11 presented below. Supplementary Fig. 10 also includes MPC1 and MPC2 control data acquired as part of this project. As described in the Reporting Summary, all antibodies used in this study had been extensively validated by members of the dedicated Histopathology/ISH Core Facility and have been since used routinely for various projects.

Supplementary Fig. 11. Example images of the LDHA and LDHB RNAscope probe validation conducted in human kidney and breast tissue samples. Scanned images of FFPE specimens from the two validation experiments are provided. In both experiments, the negative control slides had DapB in every channel, while the positive control slides included POLR2A and PPIB probes. In panel a, fluorescent scanning was performed using Akoya Phenolmager HT, with subsequent images unmixed on InForm software to separate autofluorescence. In panel b, images were acquired using Zeiss widefield microscope, with the bottom sub-panel demonstrating the outputs of a classifier used to filter out autofluorescence.

Reviewer #2 (Remarks on code availability):
metabolic_classifier.ipynb files should also be provided in GitHub instead of an attachment here.

These files can now be accessed via the following link:

https://github.com/AstraZeneca/metabolic_classifier This link has now been included in the amended Code availability section:

The code to build a metabolic tissue classifier is available at (https://github.com/AstraZeneca/metabolic_classifier). The code to conduct metabolic pathway enrichment analysis is available at (<https://github.com/AleksZakirov/MPE-analysis-for-prostate-cancer-study>)

REVIEWERS' COMMENTS

Reviewer #2 (Remarks to the Author):

The authors provided a few validation datasets to further demonstrate the rigor of the work.

Reviewer #2 (Remarks on code availability):

Please make sure the code is error-free from red highlighted regions. It makes it hard to follow with so many red highlighted errors.

Metabolic imaging across scales identifies distinct prostate cancer phenotypes
(NCOMMS-23-46894C) by Sushentsev, N. et al. (2023)

REVIEWER COMMENTS

Reviewer #2 (Remarks to the Author):

The authors provided a few validation datasets to further demonstrate the rigor of the work.

We thank the Reviewer for their feedback.

Reviewer #2 (Remarks on code availability):

Please make sure the code is error-free from red highlighted regions. It makes it hard to follow with so many red highlighted errors.

We can confirm that both codes are error-free and ready to run.